# Analysis of Global Three-Dimensional Aerosol Structure with Spectral Radiance Matching

Dong Liu[1*], Sijie Chen[1], Chonghui Cheng[1], Howard W. Barker[2], Changzhe Dong[3], Ju Ke[1], Shuaibo Wang[1], Zhuofan Zheng[1]

[1] State Key Laboratory of Modern Optical Instrumentation, College of Optical Science and Engineering, Zhejiang University, Hangzhou, Zhejiang 310027, China
[2] Environment and Climate Change Canada, Toronto, ON, Canada
[3] Shanghai Institute of Satellite Engineering, Shanghai, 201109, China

*Correspondence to*: liudongopt@zju.edu.cn

**Abstract.** A method is assessed which expands aerosol vertical profiles inferred from nadir-pointing lidars to cross-track locations next to nadir columns. This is achieved via matching of passive radiances at off-nadir locations with their counterparts that are collocated with lidar data. This spectral radiance matching (SRM) method is tested using profiles inferred from CALIPSO lidar observations and collocated MODIS passive imagery for the periods 10-25 April and 14-29 September 2015. CALIPSO profiles are expanded out to 100 km on both sides of the daytime ground-track. Reliability of constructed profiles that are removed from the ground-track by $N$ km are tested by requiring the algorithm to reconstruct profiles using only profiles that are removed from it along-track by more than $N$ km. When sufficient numbers of pixels/columns are available, the SRM method can correctly match ~75% and ~68% of aerosol vertical structure at distances of 30 km and 100 km from the ground-track, respectively. The construction algorithm is applied to the east coast of Asia during spring 2015. Vertical distributions of different aerosol subtypes indicate that the region was dominated by dust and polluted dust transported from the continent. It is shown that atmospheric profiles and aerosol optical depths (AOD) inferred from ground-based measurements agree with those constructed by the SRM method. For profiles, the relative errors between those measured by ground-based lidar and those constructed in the surrounding area are similar to the relative errors between ground-based station and CALIPSO overpass at closest distance. For AOD, the measurements from ground-based network agree with those inferred from constructed aerosol structure, better than direct observations from CALIPSO, and close to those inferred from MODIS radiances.

## 1 Introduction

Aerosol vertical structure (AVS) plays an important role in Earth's climate system. Aerosols affect changes in radiative fluxes by scattering and absorbing solar radiation, as well as modifying cloud physical properties (IPCC, 2013). Studies of Saharan dust transport (Guerrero-Rascado et al., 2008) and Pacific air-pollutant transport (Xu et al., 2019) indicate that AVS is a key parameter needed to evaluate the production, transport, and removal of aerosols. Considering the effect of external and internal

mixing of aerosols during this process, understanding AVS also helps improve descriptions of optical properties of aerosols. Similarly, the presence of aerosols induces diverse cloud responses by acting as condensation and ice nuclei (Breon et al., 2002; Textor et al., 2006). Therefore, detailed information of AVS is necessary to understand the vertical structure of clouds and precipitation (Guo et al., 2018).

The current understanding of AVS is limited by the amount of observations made across the globe. Traditional techniques rely on airborne campaigns to collect aerosols using filters on in-situ instruments (Moosmuller et al., 2009). Recently, however, the amount of relevant information has been boosted by the advent of satellite-based remote sensing. The passive sensor MODerate resolution Imaging Spectroradiometer (MODIS), on-board the Terra and Aqua satellites since 1999 and 2003, respectively, has provided global measurements almost daily (Platnick et al., 2003; Levy et al., 2013). MODIS data are used
routinely to infer aerosol optical depth (AOD). Such inferences, however, lack information pertaining to AVS. The inability to separate aerosols layers leads to difficulties in interpreting aerosol transport as high near-surface concentrations can overpower thinner layers transported aloft. In addition, MODIS is not well-suited to distinguish aerosol type, thereby making it difficult to study aerosol variations in time, space, and combinations of emission sources.

The development of lidar technology helped provide these vital missing piece of information. Ground-based lidar systems
have been stationed at various locations and also used in field campaigns to measure the vertical and horizontal distribution of aerosols (Welton et al., 2000; Welton et al., 2002; Badarinath et al., 2010). Ground-based lidars provide measurements on the fixed locations on timescale of minutes to hours, depending on the specific type of lidar used in the experiment. Limited by the stationary setting, ground-based lidars could not achieve true global coverage, nevertheless, network of ground-based lidars (e.g. MPL-NET, EARLINET, AD-NET) provide key insights to atmospheric study and are involved in validation of satellite
sensors (Kovacs et al., 2004; Mamouri et al., 2009; Pappalardo et al., 2010).

The Cloud-Aerosol Lidar and Infrared Pathfinder Satellite Observations (CALIPSO) satellite, launched in 2006, provides greater insight into AVS (Winker et al., 2009). CALIPSO's active sensor, the Cloud-Aerosol Lidar with Orthogonal Polarization (CALIOP), has the ability to resolve vertical structures of optically thin clouds and aerosols at the global scale. Yet, with its narrow nadir-viewing geometry, CALIPSO repeatedly samples only 0.2% of Earth every 16 days (Kahn et al.,
2008). This low frequency and small coverage makes it difficult to study regional AVS with CALIPSO products.

Collocation of passive and active sensors can, however, provide synergistic insights. The A-Train constellation, which includes Aqua and CALIPSO, has made many breakthroughs. Satellites of the A-train constellation are in a 705-km sun-synchronous polar orbit, with an equator-crossing time of about 1330h local solar time, and are in close proximity to one another. The success of the A-Train has led to plans to launch other active-passive satellites, such as the Earth, Clouds, Aerosols
and Radiation Explorer (EarthCARE) Mission (Illingworth et al., 2015). China has its own plan to launch a multi-functional observation satellite equipped with a High Spectral Resolution Lidar (HSRL), a Mie-Lidar, and a wide-swath cloud and aerosol imaging spectrometer targeting, amongst other things, AVS. Recognizing the limitations of using either passive sensors or active instruments, ideas for combining active-passive observations, both ground-based and space-borne, have been advanced and tested (Barker et al., 2011; Miller et al., 2014; Forsythe et al., 2000; Hutchison et al., 2006; Sun et al., 2016).

In the current study, global three-dimensional (3D) distributions of AVS are constructed utilizing CALIPSO and MODIS (Aqua) observations. 3D aerosol structure is constructed by selecting and substituting potential *donors* (pixels from CALIPSO profiles) to off-nadir *recipient* pixels within MODIS's swath based on similarity of their multi-spectral radiances. It is proposed that expansion of CALIPSO's aerosol vertical profile into the cross-track direction can fill gaps between CALIPSO tracks, thereby allowing real-global estimation of AVS. In addition, the construction provides reliable estimates of nearby AVS simultaneously with lidar measurements. The information of regional AVS has the potential to help understand short-term aerosol events, such as the heavy haze events frequently occur in the north-eastern China (Zhang et al., 2015). It could also provide assessment to cloud-aerosol interaction over a broader range than the lidar ground track (Chand et al., 2008).

Construction of AVS follows the method of scene construction proposed by Barker et al. (2011) for the EarthCARE mission. In essence, if a *donor* and a *recipient* pixel have sufficiently similar radiances, their vertical structures and column properties of clouds and aerosols are also assumed to be similar, implying that the *donor's* properties can be assigned to the *recipient*. This method, referred to here as spectral radiance matching (SRM), has been tested with respect to clouds. Barker et al. (2011) constructed 3D distributions of clouds and computed broadband radiative fluxes using 1D and 3D radiative transfer models. Results for ~100 km$^2$ domains showed good consistency when compared to measurements from Clouds and the Earth's Radiant Energy System (CERES) (Loeb et al., 2005; Loeb et al., 2007). The quality of active-passive retrievals was further analysed by Barker et al. (2014).

The objective of this study is to construct and analyze global 3D AVS with two 16-day repeat cycles of A-Train data from 10-25 April and 14-29 September 2015. AVS distributions from two seasons are compared to MODIS and CALIPSO quantities. This gives an indication of how use of MODIS-only or CALIPSO-only data might be affected by gaps in observations. To test the reliability of the constructed AVS, profiles along-track are *reconstructed* based on the same algorithm used for *construction*. The matching rate between reconstructed and measured profiles provide an approximation of the success of SRM method for aerosols.

This paper is organized as follows. Section 2 provides a brief background on the datasets used here. Section 3 reviews the SRM method, including construction and reconstruction algorithms. Section 4 assesses global construction results for two 16-day repeat cycles. The last subsection presents a case study of 3-month observations along the east coast of Asia. AOD and occurrence frequency of aerosol subtypes are analysed. Section 5 provides a summary as well as commentary on limitations of the scene construction algorithm for aerosols and scope for future applications.

## 2 Data

The SRM method for atmosphere scene construction is based on cloud and aerosol properties synergistically retrieved from active and passive sensors. In this study, we use data from CALIPSO and Aqua satellites, which make observations close in space and time (Savtchenko et al., 2007). Before CALIPSO exited the A-Train on September 13, 2018, it was flying about 73 seconds behind Aqua with its MODIS. Due to sun-glint, CALIPSO was positioned 215 km to the anti-solar side of Aqua's

ground-track at the ascending node (vice-versa on the descending node), but the wide swaths of MODIS guarantee constant collocation. Hereinafter, unless stated otherwise, "ground-track" refers to CALIPSO's.

## 2.1 CALIPSO product and MODIS product

CALIPSO carries the three-channel elastic backscattering Cloud-Aerosol Lidar with Orthogonal Polarization (CALIOP)
with passive infrared and visible imagers (Winker et al., 2002). CALIOP observes the vertical and horizontal distribution of cloud and aerosol layers, which are reported in the Level 2 vertical feature mask (VFM) product (Vaughan et al., 2009). VFM products are recorded in nominal increments of 15 consecutive laser pulses, which is equivalent to a distance of 5 km along-track. Vertical resolution of the VFM product varies from 30 to 300 m (Hunt et al., 2009;Winker et al., 2010), and it is stored as a sequence of feature classification flags with 5515 element arrays (i.e., as an N x 5515 matrix, where N is the number of
pixels recorded in the file). Each array is identified as either: clear air (1), cloud (2), tropospheric aerosol (3), stratospheric aerosol (4), surface (5), subsurface (6), no signal (7) or invalid (0 stands for bad or missing data). In this study, the VFM product was treated as the "true" indicator of atmosphere structure. This means that uncertainties in the product propagate to analyses of results derived from it. To minimize this impact, only arrays identified with high confidence - cloud-aerosol discrimination (CAD) scores larger than 70 - were used. In addition, attention was paid to extinction quality assurance (QA)
flags. For most constructed scenes, only aerosols layers with a QA flag of 0 and 1 were included. This helped avoid large errors that can stem from the nonlinear behaviour of the AOD retrieval (Huang et al., 2015a). CALIPSO products used in the work are from Version 4.20.

The MODIS passive sensor has 36 channels spanning visible to thermal wavelengths. MODIS Level 1 products contain calibrated radiances at 1 km resolution (MYD021KM) and are used to infer several key properties of clouds and aerosols
(Kaufman et al., 2002; Minnis et al., 2008; Levy et al., 2013; Platnick et al., 2017). Pixel locations and ancillary information (MYD03) are used by the algorithm discussed in the next section. Because MODIS retrieval errors increase with solar and viewing zenith angles (Kato and Marshak, 2009), analyses were restricted to between latitudes 60 ° N and 60 ° S. MODIS products used in the work are from MODIS Collection 6.

MODIS's wide swath (2330 km cross-track) ensures collocated observations with CALIPSO. Their narrow collocation
track is referred to as the active-passive retrieved cross-section (RXS) (cf. Barker et al. (2011)). In the following sections, the actual RXS is referred as "RXS-nadir", while the RXS as expanded by the SRM method, up to 100 km on both sides of the RXS-nadir, is referred to as "RXS-expand".

As CALIPSO products are provided at 5 km resolution, and MODIS products at 1 km, they were merged using a grid formed by latitude and longitude of the first and the last laser shot with cross-track width doubled to ensure the grid is larger
than 1 km. Radiances, solar zenith angle, and solar azimuth angle geolocated within each grid were averaged. In terms of surface type, if one grid contains both land and sea flag from MODIS, it was redefined as a mixed surface type. The same grid size was applied to the RXS-expend.

## 3 Method

### 3.1 Spectral radiance matching (SRM) method

Barker et al. (2011) proposed a scene construction algorithm to extend cloud-aerosol profiles in the RXS-nadir to nearby off-nadir positions. They also provided a *reconstruction* algorithm to assess the *construction* algorithm's performance. The construction algorithm is based on matching spectral radiances of nadir pixels with those of off-nadir pixels, while the reconstruction algorithm mimics the process by setting a *dead zone* around RXS pixels and filling them with other RXS pixels that reside outside the dead zone. The method is reiterated briefly here for the convenience of readers.

To match and substitute the most suitable *donor* at location $(m,0)$ on the RXS for each off-nadir *recipient* at location $(i,j)$ in the passive swath $j \in [-J, -1] \cup [1, J]$, the SRM method computes a cost function $F(i,j;m)$ during daytime as

$$F(i, j; m) = \sum_{k=1}^{K=4} \left( \frac{r_k(i, j) - r_k(m, 0)}{r_k(i, j)} \right)^2 ; m \in [i - m_1, i + m_2], \tag{1}$$

where $r_k$ is MODIS radiance for the $k^{th}$ band, $m \in [i - m_1 \cup i + m_2]$ is the range of potential donors along the RXS. Potential donors also need to satisfy: 1) the same underlying surface type as the recipient; 2) similar solar zenith and solar azimuth angles as at the recipient; 3) |CAD| $\geq 70$. See Barker et al. (2011) for details explaining each condition. For this study, $K = 4$ was used (0.62-0.67, 2.105-2.155, 8.4-8.7, and 11.77-12.27μm). The bands are chosen for their widely accepted usage in retrieving cloud properties, including cloud cover, cloud top properties (CTP/CTT/CTH), and cloud phase (Ackerman et al., 1998; Baum et al., 2012; Baum et al., 2000) as well as aerosol properties (Sayer et al., 2014; Levy et al., 2013; Remer et al., 2013). The possible combinations of different bands have been tested and details are provided in Section 3.2.

To ensure the number of potential donors for far off-nadir recipients, search range was adopted from Sun et al. (2016) and defined as

$$m_1 = m_2 = \left\{ \begin{array}{l} 200; \cdots\cdots\cdots D_m \leq 30 \\ 200 + D_m; \cdots D_m > 30 \end{array} \right. , \tag{2}$$

where $D_m$ is the shortest distance between recipient and RXS. $F(i,j;m)$ is then ordered from smallest to largest, and the Euclidean distance between a potential donor at $(m,0)$ and the recipient at $(i,j)$ is calculated as

$$D(i, j; m) = \Delta L \sqrt{(i - m)^2 + j^2} , \tag{3}$$

where $\Delta L$ is horizontal resolution (1 km) of MODIS radiance measurements. The most suitable donor is found by solving

$$\underset{m^* \in [1, (m_1 + m_2 + 1)f]}{\arg\min} \{D(i, j; m^*)\}; f \in (0, 1) , \tag{4}$$

which means, the selected donor, noted with asterisk $(m^*, 0)$, is closest to the recipient and has sufficiently similar radiances. $f = 0.15$ was used here rather than 0.03 as in Barker et al. (2011),. This is because CALIPSO aerosol products have lower resolution (5 km) than cloud products (1 km).

The reconstruction algorithm, on the other hand, is designed to evaluate the performance of the construction algorithm, since the reconstructed results along the RXS can be compared to the actual observations. A dead zone centred at the recipient $(i,0)$ is set by defining the selection range for potential donors as $[i-m_1, i-n] \cup [i+n, i+m_2]$. By barring selections of potential donors from the nearest $\pm n$ pixels, the reconstruction process is forced to resemble the filling of an off-nadir recipient removed from the RXS by $n$ pixels. The results of reconstruction thus give an approximate indication of how well the SRM method can be expected to perform.

## 3.2 Selection of bands for aerosol applications

Since the construction algorithm is initially developed for clouds, efforts have been made to apply the algorithm to aerosols. The following test is performed to find the possible combination of bands most sensitive to aerosols. 30 days of CALIPSO profiles at the east coast of China in 2015 are selected and screened with clear-sky and heavy aerosol loadings conditions. The manually selected cloudless datasets with heavy loading events are expected to give a clear indication of whether the algorithm could work for aerosols or not. The following combinations of radiance bands are tested: 1) a combination of bands 1, 7, 29 and 32 used by Barker et al. (2011); 2) a combination of bands 1 and 7 only; 3) a combination of visible bands 1, 2, 3 and 4. The performance of algorithm using these combinations is evaluated by reconstructing the profile with dead zone setting for 30 and 100 km.

A typical comparison among the reconstructed profiles is shown in the Fig. 1, where the panel (a) is the original profile, panel (b) to (d) corresponds to combination 1 to 3 described above, panel (e) shows the reconstructed profile from directly choosing the closest pixels outside the dead zone. The results of the test indicate that the bands used by Barker et al. (2011) could get a pretty successful reconstruction. The matching rate at 30 and 100 km are on average 81.9% and 75.2%, respectively, which means this combination can be used to construct aerosol vertical structure. In contrast, using visible bands only have lower matching rate (around 60-70%), especially when aloft aerosol layer are present. Selecting the closest pixel, on the other hand, has very high matching rate at 30 km, which is expected since aerosol properties are relatively horizontally uniform. However, as the dead zone range increases or in cases that aerosol layer is not continuous, the simple horizontal shift leads to more errors. Based on the test results with heavy aerosol loading events, the combination of bands for aerosol application is the same as the wavelength selection in Barker et al. (2011).

## 3.3 Theoretical best matching (TBM) method

Differences between reconstructed profiles and CALIPSO observations can have two main causes: 1) the SRM method does not select the best matching donor from the potential donors; and 2) the profiles for the actual best matching donor and the recipient differ. To analyse the contribution of these two causes, a *theoretical best matching* (TBM) method was devised to purposely select the most suitable donor from the potentials. In theory, the result from TBM method is the best that can be expected from the SRM method as defined. If for every column along the ground-track, there exists an identical column within range but outside the dead zone, the TBM method would reconstruct the original profiles perfectly. Therefore, differences

between the TBM method's reconstructed profiles and original profiles indicate the influence of the second cause. Differences between reconstructed profiles from the TBM method and constructed profiles from the SRM method, on the other hand, indicate the influence of the first cause.

The TBM method of reconstruction is calculated by comparing CALIPSO's VFM for each potential donor (constrained by *dead zone*) to that of the recipient. The comparison between reconstructed results and a VFM is categorized as shown in Table 1. Results of each array are classified as either 'agree' or 'disagree' for clear (1), cloud (2), or aerosol (3 = tropospheric, 4 = stratospheric). Recipient arrays that are identified as 'no signal' are (7) and 'invalid' (0), and are not counted when comparing with possible donors as the actual scenes at these elements are unknown. Recipient arrays that are identified as surface (5) or subsurface (6) are also not counted as matches of these two feature types are considered less important. On the other hand, arrays classified as no signal, invalid, surface, or subsurface for potential donors are counted as 'disagree' when comparing to the *recipient*. Hence, a matching rate can be calculated as

$$\text{MR} = (\text{Agree}_{cr} + \text{Agree}_{cd} + \text{Agree}_{ae}) / N , \tag{5}$$

where $N$ is the total number of VFM arrays measured along the ground track that are identified as clear air, clouds or aerosols. The potential donor with largest MR is selected as the donor for the TBM method.

## 4 Results and Discussion

Results are presented in three subsections. The first two diagnose the construction algorithm while the third employees it specifically to the east coast of Asia.

### 4.1 Expansion of active-passive retrieved cross-section (RXS)

This section presents results of constructed and reconstructed aerosol properties using two full CALIPSO 16-day repeat cycles in 2015, and a comparison of the TBM results against original observations. From 10 - 24 April and 14 - 29 September, CALIPSO functioned normally except during a boresight diagnostic and alignment on 18 September, losing about half of that day's data. Because MODIS retrievals of aerosol properties depend mainly on visible wavelengths, only daytime observations were used in this study.

Table 2 summarizes frequencies of occurrences of atmosphere conditions. These numbers refer to the occurrence of atmospheric features as percentage they occupied in the vertical column. We calculated this occurrence rate according to CALIPSO VFM products, which was then scaled for vertical and horizontal resolution of the products (Hunt et al., 2009).The majority of conditions were identified as clear. Above 8.2 km clear arrays occur over 90% of the time. Aerosols and clouds occurred below 8.2 km, in about 7% and 5% of the cells, respectively. Note that arrays identified as 'no signal' represent 16 - 18% of cells in this layer; CALIPSO's signal can be totally attenuated beneath opaque clouds and certain aerosols. This indicates that the numbers in Table 2 likely underestimated the amount of clouds and aerosols in actual atmosphere. After removing elements identified as 'no signal', surface, and sub-surface, aerosols and clouds occupied 4.43 - 4.52% and 5.35 -

6.15% of the cells; clear-skies account for the remainder. The horizontal cloud coverage between $60\,^{\circ}$N and $60\,^{\circ}$S for the tested periods in April and September 2015 are 68.7% and 71.3%, respectively.

The RXS is expanded to 100 km on both sides of it by constructing profiles along 40 parallel tracks every 5 km. Figure 2 shows an example of a CALIPSO track passing the African coast on 23 April 2015 between $5\,^{\circ}$S and $15\,^{\circ}$N. For better visualization, only 4 extended tracks are shown on both sides of the CALIPSO track, each separated by 25 km. The clear arrays on the extended tracks are made transparent.

Height-resolved global AOD maps (averaged for a $2\,^{\circ} \times 2\,^{\circ}$ lat/long grid) based on the two selected periods are shown in Fig. 3. In the near-surface layer, 2 km above ground level (AGL), in April, relatively high aerosol loadings are found in the cross-Atlantic African dust transport, Saudi Arabia, and India. In September, dust dynamics are much weaker but much biomass burning is apparent in the Brazilian Amazon and Southern Africa. This seasonal trend of dust and smoke is more obvious in the layer 2-4 km AGL. Aerosol in this layer aloft are expected to be undergoing long-range transport. In April, the thickest dust layers are found slightly inland of the western coast of Africa, around 12.5 °N, 5.5 °E, and in the centre of Saudi Arabia around 24.5 °N, 42.5 °E. The shift of AOD distribution between surface layer and layer above is logical, and indicates the movement of dust layers as the aerosol loadings are transported towards the oceans. In September, this contrast is harder to observe as the dust dynamic is weaker, but similar trends are found in the biomass burning regions. In addition, persistent high aerosol loadings in both 0-2 km and 2-4 km AGL are found in India and the east coast of China with mixed sources of natural aerosols and pollutants. The results could be affected by the local topography. Marine aerosols are confined largely to the near-surface layer, with some vertical transport in Southeast Asia in September due to the Asian monsoon. The observed pattern is mostly consistent with other studies in terms of global distribution and seasonal variations (Martins et al., 2018; Liu et al., 2012; Chen et al., 2018).In comparison, globally-averaged AOD from RXS-expand is 0.0027 larger than observations made by CALIPSO in April, and 0.0028 larger in September. Both positive and negative differences in regions with high aerosol loadings, but none of the regions exhibit consistent high or low biases. Therefore, these insignificant differences are likely caused by random errors in the algorithm in conjunction with CALIPSO's unbiased sampling, and suggest that aerosol distributions constructed using SRM method do not change the global aerosol mass as inferred directly from CALIPSO data.

**4.2 Reconstruction of RXS-nadir**

To evaluate the reliability of its result, RXS cross-sections were reconstructed for *dead zones* set to 30 km and 100 km, which should give an approximate evaluation of the chances of successfully constructing scenes nearby and well removed from the RXS. Detailed analysis of matching rate between reconstructed RXS and RXS-nadir is summarized in Table 3.

During each 16-day cycle, there were about 600,000 CALIPSO observations made in the selection range. With *dead zone* set to 30 km, about 95% of them, as recipients, are matched up with selected donors. The remaining 5% are not matched because no suitable donors are found in the range with conditions described in Section.3. This ratio increases with the distance of *dead zone*, because the number of potential donors that meet the requirement of surface type and solar angles decreases with

increasing distance between donor and recipient. When *dead zone* is set to 100 km, only about 71% of recipients are matched with a donor. The following analysis focuses on the portion of recipients matched up with donors, unless mentioned otherwise.

Overall, at 30 km, the reconstruction based on SRM method correctly matches 92.04% of air columns in April, and 92.55% in September. At 100 km, SRM method correctly matches 88.57% of air columns in April, and 89.68% in September. To investigate the reasons behind imperfect reconstruction, the correctly reconstructed arrays and incorrectly reconstructed arrays are analysed separately (see Table 3).

The same analysis is also performed with the TBM reconstruction in which 96.87% of air column is correctly reconstructed in April, and 92.55% in September at 30 km. At 100 km, 93.95% of air column is correctly reconstructed in April, and 94.76% in September. As discussed in Section 3.2, the difference between TBM reconstruction and perfect reconstruction (i.e. 100% correct reconstruction of nadir profiles) indicates the errors caused by selection from limited numbers of donors.

On the other hand, differences between TBM reconstruction and SRM reconstruction indicate that the SRM method still needs improvement. In comparison, over 50% of the mismatches by TBM reconstruction come from clear arrays; higher than that from SRM reconstruction. The fraction of no signal (5-9%) in the mismatch of the TBM reconstruction is much lower than that of SRM reconstruction. This results partly from the procedure of the TBM method which tends to choose donors with less arrays of no signal. The fractions of mismatch of clouds and aerosols are not significantly different between the methods.

Since the analysis of the entire air column is easily overwhelmed by clear arrays, the matching rate with respect to aerosols is calculated. The ratio is obtained as the number of correctly reconstructed aerosol arrays divided by the total number of aerosol arrays in the original profile (both correctly and incorrectly matched in the reconstructed profile), and other arrays mismatch as aerosols in the reconstructed profile.

The average matching rate with respect to aerosols across the globe is 68.18% at 30 km, and 62.33% at 100 km. The matching rate is higher over land than over ocean, possibly because there are more aerosols over land. The matching rate also shows a general trend with latitude. At 30 km, the average matching rate is 73.78% between 14 °-24 °N, 71.79% between 14 °-24 °S, and 66.92% between 4 °N - 4 °S. At 100 km, the average matching rate is 67.43% between 14 °-24 °N, 66.18% between 14 °-24 °S, and 59.91% between 4 °N – 4 °S. This is linked to the persistent high cloud fraction in the Intertropical Convergence Zone: as clouds attenuate CALIPSO's signal, the ratio of mismatching in the reconstruction increases.

In addition, the matching rate is strongly affected by the number of observed pixels (containing aerosols) in the grid, especially at high latitudes. Figure 4 contains a box plot analysis which indicates that sample variance is high for grids lacking sufficient data points (Fig. 4). The boxes are separated by the number of pixels in 1 °x 1 °grid. Boundaries of each box represent the 25[th] and 75[th] percentiles of the sample data in the grid, the central red line marks medians, the length of whiskers corresponds to approximately +/-2.7σ and 99.3 percent coverage assuming the data are distributed normally. At both distances, the matching rate of aerosol increases steadily while the span of data decreases with the number of pixels in the grid. For grids with more than 20 pixels over the two CALIPSO cycle, the average matching rate of aerosol is 75.32% at 30 km, and 68.52%

at 100 km. This could be explained by the fact that in regions where aerosols occur more frequently (thus have more pixels observed), suitable donors are easier to find.

## 4.3 Case study

With the complete datasets of CALIOP profiles, MODIS radiances and geolocation fields, construction based on the SRM method can be applied worldwide. It is applied here to aerosols along the east coast of Asia (117 °-132 °E, 26 °-41 °N) for a 3-month period (MAM) in 2015. SRM method's AOD estimates are compared to AErosol RObotic NETwork (AERONET) values inferred from ground-based sun-photometers on a day-by-day basis.

The east coast of Asia represents one of the most complicated aerosol regions as it includes transported natural dust, anthropogenic dust, black carbon (BC) and organic carbon (OC) from biomass burning, as well as mixtures of BC, OC and sulfates from urban pollution (Logan et al., 2013; Huang et al., 2015b). Analysis of multiple AERONET sites in East Asia during the 2001-2010 period showed that the area is dominated by mineral dust during spring months, likely transported from the Gobi and Taklamakan Deserts (Eck et al., 2005; Huang et al., 2008; Logan et al., 2013). The dominance of these outflowing aerosols continues to be observed as far away as Japan (Ikeda et al., 2014; Uchino et al., 2017).

Figure 5 shows seasonal distributions of AOD in the area. Data are binned in a 0.25 ° x 0.25 ° lat/long grid. A couple locations with seasonal-mean AOD > 0.5 occur on the main land of China, surrounding the Bohai Sea, and the Sea of Japan. Large AOD across the Bohai Sea and near the island of Japan indicate mass-transportation of aerosols during this season. A region of small AOD (< 0.1) exists to the east of Shanghai, which might be caused by frequent failures of satellite retrievals at the Yangtze River Delta and the Yellow Sea due to turbidity of local water. Regional distribution based on MODIS AOD products shows certain similarity, but is higher in general. Note that the extreme high values (AOD > 1) over the Yellow Sea and inside the Bohai Bay (i.e., the same area where RXS-expand shows small AOD values), result from few measurements; 10 times less than other cells. The small sample sizes from both CALIPSO and MODIS suggest the large difference here is due to difficult to handle local surface conditions.

To better analyse the source of aerosols, the area was divided into three regions along two CALIPSO ground-tracks (Fig. 5). Region A mainly includes land inside China. Region B includes inshore coastal waters between China and the Korean peninsula, as well as part of the East China and Yellow Seas. It also includes the most populated area of South Korea. Region C includes the remaining area of the Korean peninsula, Kyushu island of Japan, and surrounding waters.

Vertical distributions of aerosol subtypes for these regions are shown in Fig. 6. In Region A, the average extinction profile indicates aerosol layers between 1-2 km, near 3.7 km, and above 5 km. The two lower layers are dominated by polluted dust, while the upper one is mainly clean dust. Occurrence of polluted continental aerosols and polluted dust suggests local aerosol production. The increase of smoke near 2.8 km suggests some transportation of biomass burning aerosols, possibly due to spring agricultural practices and indoor heating. The average extinction profile for Region B is smaller than that for Region A. These aerosols are dominated by polluted dust below 2 km, and clean dust above. The distribution of polluted dust and dust could be explained as transported from Region A with decrease in altitude. There might be some transport of polluted

continental aerosols or dusty marine aerosols, but no transport of smoke is observed. In Region C, the average extinction profile shows few aerosols above 2 km except a thin layer at 5.5 km. Near surface layer is composed of dust, dusty marine, clean marine, and polluted continental aerosols. The fraction of clean marine is highest among the three regions, possibly due to large ocean area and major harbours in Busan and Kyushu. The upper layer of dust and polluted dust is questionable at first glance. Detailed analysis showed that the regional AOD, with especially large values near Japan, is caused by high values on April 17, 2015. Two days before, in the afternoon of April 15, 2015, China recorded the most severe dust storm since 2002 across Beijing, known as the "4.15 Dust Storm". Large values of AOD near Japan might be due to the influence of this major aerosol transport event, while the middle layer might be missed due to limited viewing by CALIPSO. In fact, a small region of relatively high AOD near Japan is also shown in Fig. 5 for MODIS.

In the three-month period, the Asian dust and aerosol lidar observation network (AD-Net) site at Seoul, Korea (37.5N,127.0E) provided measurements of atmospheric profiles that we were able to compare with those constructed in the surrounding area using the SRM method. Seoul station has a standard lidar system in AD-Net, which is a two-wavelength (1064 nm, 532 nm) polarization sensitive (532 nm) Mie-scattering lidar, plus a 532 nm Raman (Shimizu et al., 2004). Based on the ground track, the A-Train sensors made overpass near the station for a total of 6 days during that spring. However, 4 out of these 6 days were heavily cloudy. For the remaining 2 days, 7 March and 24 April, the comparisons among ground-based lidar profiles, CALIPSO profiles at shortest distance and RXS-expand profiles averaged 25 km around the location of Seoul station are shown Fig. 7.

The CALIPSO measurements used for comparisons are level 1.5 data products of attenuated backscatter profiles, which clouds, overcast, surface, subsurface, and totally attenuated samples have been removed before being averaged to a 20 km horizontal resolution. In this case, RXS-expand profiles are based on the same products. The ground-based measurements used for comparison are the 532 nm attenuated aerosol backscatter coefficient products, averaged within 2 hr before and after the satellite overpass with 15 min time resolution.

For the aerosol layer 0-4 km above the ground, the relative error between CALIPSO profiles and ground station profiles are on average 21.6% on 7 March, and 18.7% on 24 April. The distances between station and ground track are 51.0 km on the first day, and 50.1 km on the second. Between RXS-expand profiles and ground station profiles, the average relative errors are 27.9% and 23.4%, respectively. The results from the comparisons agreed in general. Previous studies found that there were considerable disagreement between CALIPSO measurements and ground-based lidar measurements; in most studies, the differences were found to be around 20% (Mamouri et al., 2009; Wu et al., 2011; Kim et al., 2008; Chiang et al., 2011).

In the same period, five AERONET sites in the selected region recorded 20 comparable measurements between RXS-expand and AERONET, with the CALIPSO ground-track passing within 100 km of AERONET sites (see Fig. 8). AERONET AOD at 500nm were constrained to ±2 hours of CALIPSO's passing. The RXS-expand AOD at 532nm is averaged using the 10 pixels closest to each AERONET site. Comparisons were made between RXS-expand AOD and AERONET AOD, as well as RXS-nadir AOD and AERONET AOD. Among these collocated measurements, 17 measurements (85%) show better agreements between AERONET and RXS-expand values of AOD. One outlier (grey circle) was found for Baengnyeong on

May 21$^{st}$. Measurements from Baengnyeong have a larger span than the other sites in general, which might be explained by strong sea winds across the Bohai Sea which often change direction and speed and may lead to large variations in aerosol transport. Removing the outlier, the correlation between RXS-expand AOD and AERONET AOD is R = 0.88, whereas for RXS-nadir AOD and AERONET AOD it is R = 0.80. Still, however, both are less than R = 0.95 between MODIS AOD and AERONET AOD.

## 5 Summary

Three-dimensional aerosol structure is constructed across the globe using vertical profiles from CALIPSO and MODIS radiances. Based on matching and substituting nadir pixels (*donors*) into off-nadir pixels (*recipients*) with similar radiances, the atmosphere's vertical structure is expanded up to 100 km on both sides of the ground-track. The construction results fill gaps between CALIPSO ground-tracks and increase the frequency of observations for some areas by as much as once in 8 days as opposed to CALIPSO's 16 days. Consequently, the construction algorithm approximates aerosol vertical structure at locations never measured by CALIPSO; this has the potential to improve our understanding of regional distributions of aerosols. Increasing the number of observations can also help reduce the CALIPSO-centric selection bias by allowing for the study of aerosols over short time periods and small regions.

Reconstruction of nadir profiles verifies the overall performance of using the SRM method to construct 3D cloud-aerosol structure as a function of distance from ground-track. By mimicking off-nadir distance with a *dead zone* along ground-track profiles, reconstruction of nadir profiles shows that at 30 km from the RXS the SRM method correctly matches 92% of cells in April and September 2015. At 100 km, these rates drop to ~89%. Comparison to TBM reconstruction suggests that the limited number of columns available from the active sensor is responsible for ~3% of mismatches at 30 km and ~6% at 100 km. Differences between profiles reconstructed using SRM and TBM methods come mostly from profiles whose lidar signals are totally attenuated. Otherwise, the fraction of mismatching of aerosols or clouds is similar. With a sufficient number of observations, the average matching rate of aerosol could reach 75% at 30 km, and 68% at 100 km.

The construction algorithm was applied to the east coast of Asia where heavy aerosol loadings are frequent. The expansion of RXS successfully provided regional distributions of aerosols in spring 2015. Analysis of vertical distributions of each aerosol subtype showed that regional west-east transportation was dominated by dust and polluted dust. Comparison of atmospheric profiles against AD-Net lidar station shows agreement with constructed profiles and CALIPSO measurements at similar level. Comparison of column AOD against ground-based AERONET values shows better agreement with expanded RXS relative to CALIPSO observations. The correlation increases from R = 0.80 to R = 0.88.

The construction of 3D aerosol structure based on the SRM method appears as though it could be an important tool to analyse global and regional aerosol properties. Though the matching rate is not perfect, improvements to the algorithm seem likely. The method in this work is not intended to get a precise quantification of aerosol profile, but to provide an estimate of the column's vertical structure. We did expect, to some extent, the estimation could be improved through calculations with

constrains such as the column AOD measured by passive sensor at the exact location of the recipient pixel, which will need a lot more work in the future. In addition, since lacking of more suitable pixels is responsible for about half of the mismatching results, we are looking forward to launching more satellites with active and passive sensors, and possibly combining data from multiple satellite systems.

*Data availability:* NASA EOSDIS Land Processes Distributed Active Archive Centre (LP DAAC) provided MODIS data (available at: https://lpdaac.usgs.gov/data_access/data_pool, LP DAAC, 2019). NASA Langley Research Centre (LaRC) provided CALIPSO data (available at: https://subset.larc.nasa.gov/calipso/, LaRC, 2019).

*Author contribution:* Sijie Chen designed the experiments and Chonghui Cheng helped carried them out. Professor Barker

developed the scene construction method utilized in this work and gave suggestions during the progress. Dr. Dong is responsible for resources and funding acquisition. Ju Ke, Shuaibo Wang and Zhuofan Zheng helped data analysis. Professor Liu supervised the project and the preparation of the manuscript with contributions from all co-authors.
*The authors declare that they have no conflict of interest.*

*Acknowledgements.* This work was supported by National Key Research Program of China (2016YFC0200900); National Natural Science Foundation of China (NSFC) (41775023); Excellent Young Scientist Program of Zhejiang Provincial Natural Science Foundation of China (LR19D050001). We would also like to thank all the reviewers and editor for the time spent to evaluate our work and for the useful and constructive comments they gave.

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

**Table 1.** Contingency tables for the comparison of the reconstructed profile using SRM method to CALIPSO VFM product.

| Agree | Recipient | Donor | Disagree | Recipient | Donor |
|---|---|---|---|---|---|
| Agree$_{cr}$ | 1 | 1 | Disagree$_{cr-cd}$ | 1 | 2 |
| | | | Disagree$_{cr-ae}$ | 1 | 3 or 4 |
| Agree$_{cd}$ | 2 | 2 | Disagree$_{cd-cr}$ | 2 | 1 |
| | | | Disagree$_{cd-ae}$ | 2 | 3 or 4 |
| Agree$_{ae}$ | 3 | 3 | Disagree$_{ae-cr}$ | 3 or 4 | 1 |
| Agree$_{ae}$ | 4 | 4 | Disagree$_{ar-cd}$ | 3 or 4 | 2 |
| | | | Disagree$_{suf}$ | 1 - 4 | 5 or 6 |
| | | | Disagree$_{nosig}$ | 1 - 4 | 7 |

The bits are interpreted as clear air (1), cloud (2), tropospheric aerosol (3), stratospheric aerosol (4), surface (5), subsurface (6), or no signal (7).

**Table 2.** Summary of Atmosphere condition from 10 to 24 April, and from 14 to 29 September 2015.

| | Aerosol (st) | Aerosol(tr) | Clear | Cloud | No signal | Surf/Subsurf |
|---|---|---|---|---|---|---|
| 20.2 km -30.1 km | 0.14% | - | 99.84 - 99.85% | 0.01 - 0.02% | - | - |
| 8.2 km -20.2 km | 0.038% | 0.24 - 0.28% | 91.87 - 93.71% | 4.77 - 6.41% | 1.24 - 1.31% | - |
| -0.5 km -8.2 km | - | 7.08 - 7.12% | 63.42 - 65.32% | 5.48 - 5.68% | 16.48 - 18.13% | 5.64% |

**Table 3.** Summary of comparison between observed nadir profiles and reconstruct nadir profiles.

| | Match | Clear | Cloud | Aerosol | Mismatch | Clear | Cloud | Aerosol | No Sig* | Surf* |
|---|---|---|---|---|---|---|---|---|---|---|
| **Dead Zone 30 km** | | | | | | | | | | |
| **10-24 April 2015** | | | | | | | | | | |
| Actual | 522200 | 480846 | 24057 | 17297 | 45160 | 14920 | 11665 | 6688 | 11197 | 689 |
| | 100% | 92.08% | 4.61% | 3.31% | 100% | 33.04% | 25.83% | 14.81% | 24.79% | 1.53% |
| Theo | 549580 | 496553 | 31243 | 21784 | 17780 | 9298 | 4804 | 2291 | 1086 | 301 |
| | 100% | 90.35% | 5.68% | 3.96% | 100% | 52.30% | 27.02% | 12.88% | 6.11% | 1.69% |
| **14-29 September 2015** | | | | | | | | | | |
| Actual | 514340 | 478282 | 19329 | 16729 | 41380 | 13770 | 10336 | 6522 | 10018 | 733 |
| | 100% | 92.99% | 3.76% | 3.25% | 100% | 33.28% | 24.98% | 15.76% | 24.21% | 1.77% |
| Theo | 540220 | 493044 | 25818 | 21357 | 15500 | 8274 | 3944 | 2082 | 885 | 316 |
| | 100% | 91.27% | 4.78% | 3.95% | 100% | 53.38% | 25.44% | 13.43% | 5.71% | 2.04% |
| **Dead Zone 100 km** | | | | | | | | | | |
| | Match | Clear | Cloud | Aerosol | Mismatch | Clear | Cloud | Aerosol | No Sig | Surf |
| **10-24 April 2015** | | | | | | | | | | |
| Actual | 368330 | 342896 | 12891 | 12543 | 47540 | 16977 | 11378 | 7875 | 10374 | 937 |
| | 100% | 93.09% | 3.50% | 3.41% | 100% | 35.71% | 23.93% | 16.56% | 21.82% | 1.97% |
| Theo | 390690 | 357283 | 17358 | 16049 | 25180 | 13367 | 5247 | 3981 | 2066 | 520 |
| | 100% | 91.45% | 4.44% | 4.11% | 100% | 53.09% | 20.84% | 15.81% | 8.20% | 2.07% |
| **14-29 September 2015** | | | | | | | | | | |
| Actual | 382520 | 359847 | 10101 | 12572 | 44020 | 15379 | 9725 | 7793 | 10112 | 1011 |
| | 100% | 94.07% | 2.64% | 3.29% | 100% | 34.94% | 22.09% | 17.70% | 22.97% | 2.30% |
| Theo | 404170 | 373653 | 14077 | 16441 | 22370 | 11711 | 4117 | 3987 | 2017 | 538 |
| | 100% | 92.45% | 3.48% | 4.07% | 100% | 52.35% | 18.40% | 17.82% | 9.02% | 2.41% |

*'No Sig' stands for 'no signal' and 'Surf' stands for 'surface and subsurface' portions of the measured columns.

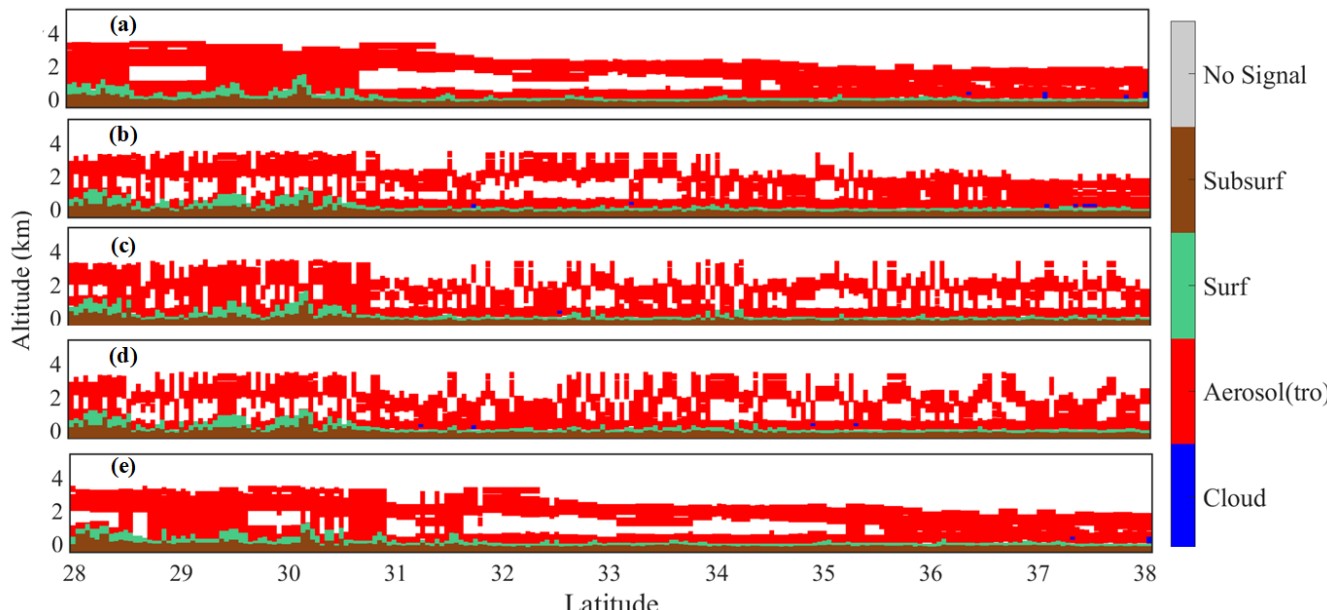

**Figure 1** Reconstruction of CALIPSO profile passing the east coast of China on 3 January 2015 with dead zone setting for 100 km. The panels show the original profile and reconstructed profiles using different combinations of radiance bands.

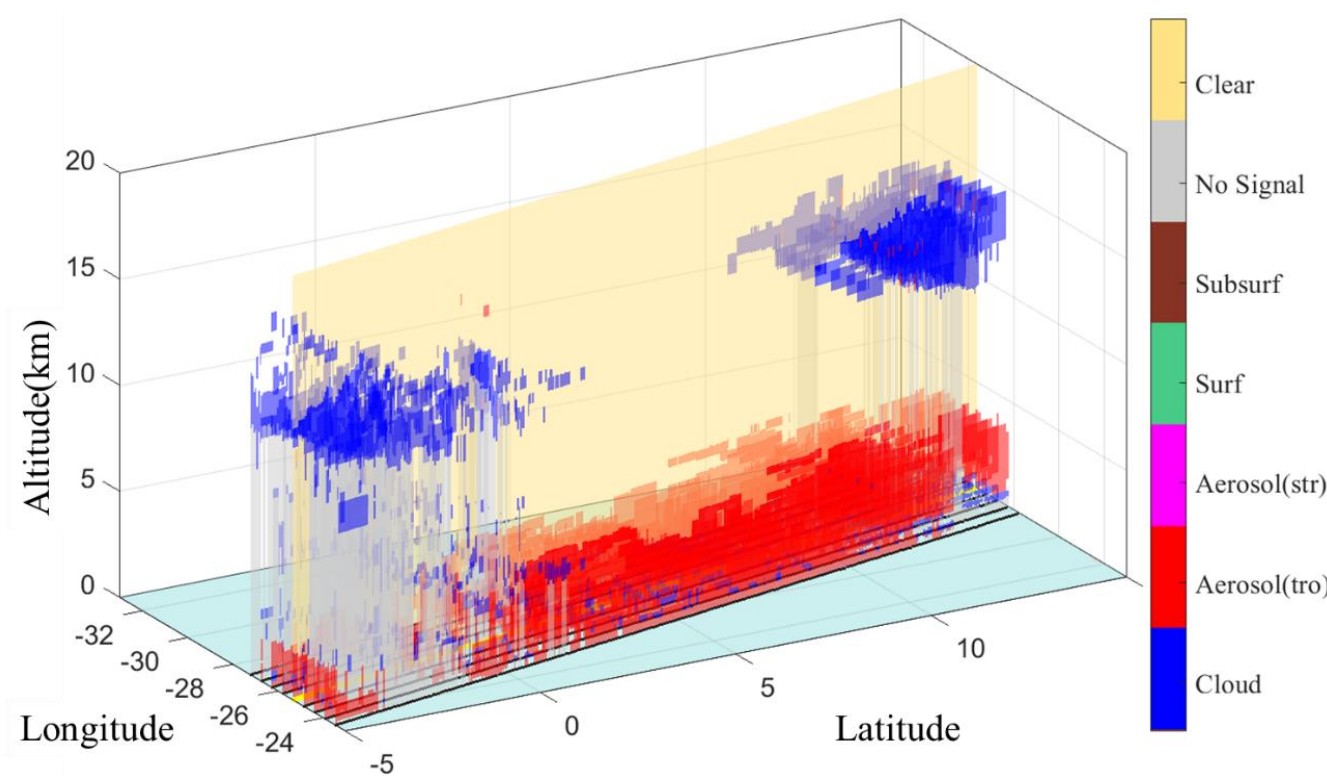

**Figure 2** Demonstration of the expansion of the active-passive retrieved cross-section (RXS), which runs through the centre, for a transect near Africa-Atlantic coast between 15 °N and 5 °S on 23 April 2015.

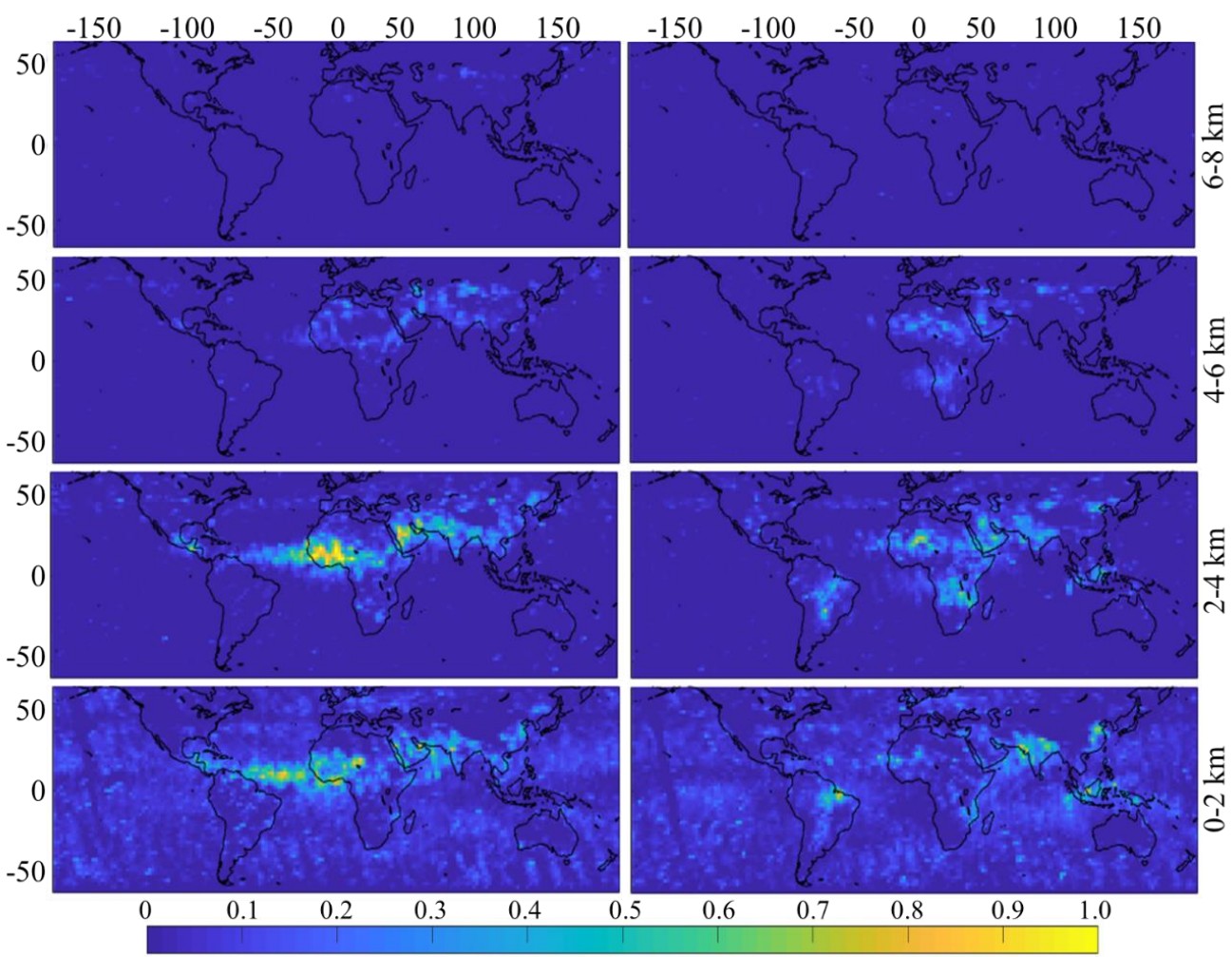

**Figure 3** Height-resolved global distributions of aerosol optical depth (AOD) based on construction of RXS expanded 100 km on both sides of CALIPSO's ground-track. Left column is for the April dataset and the right column is for September . Data are binned on a 2 °lat/long grid.

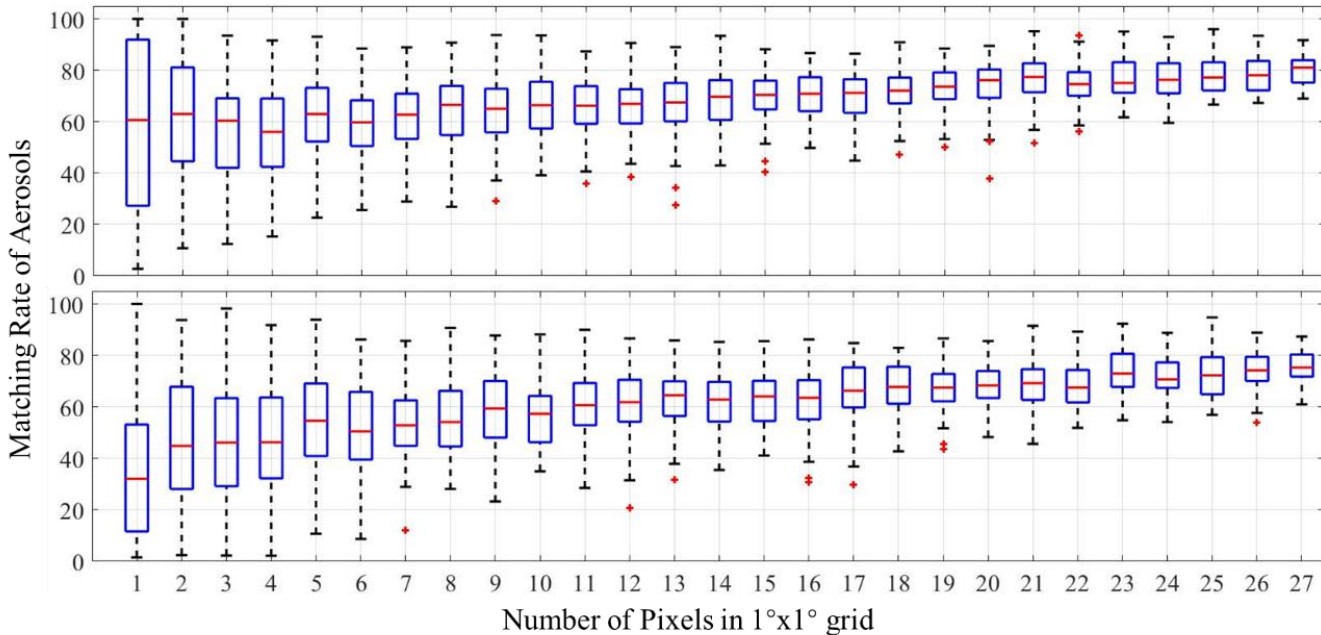

**Figure 4** Matching rates of aerosols as a function of number of samples in 1° grid cells for reconstructions of nadir profiles with 30 km dead zone (top) and 100 km dead zone (bottom). Boundaries of each box represent the 25th and 75th percentiles of the sample data in the grid, red lines indicate medians, and whiskers correspond to approximately +/-2.7σ (99.3 percent coverage assuming data are distributed normally). Points marked with '+' are extreme outliers.

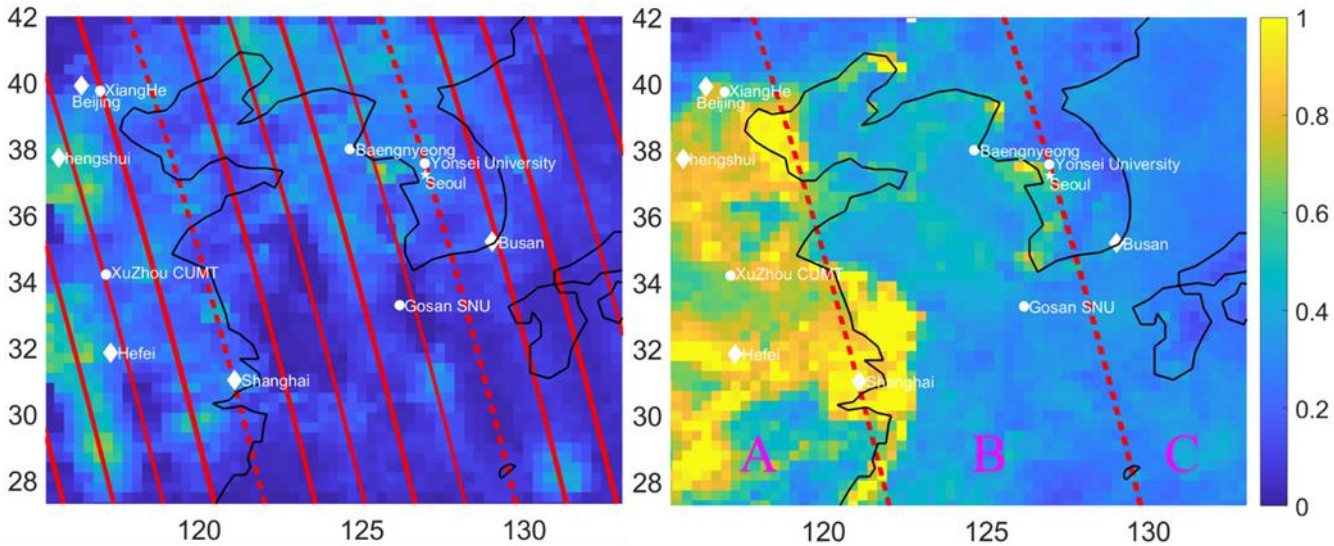

**Figure 5** Seasonal distribution of aerosol optical depth (AOD) between 117 °-132 °E, 26 °-41 °N for March to May, 2015 based on RXS-expand (left) and MODIS (right). Lines mark CALIPSO ground-tracks. Dotted tracks are used as the boundaries for the analysis of aerosol subtypes in Fig. 6. White circles mark AERONET sites. Star marks the AD-Net site at Seoul. Diamonds mark some major cities.

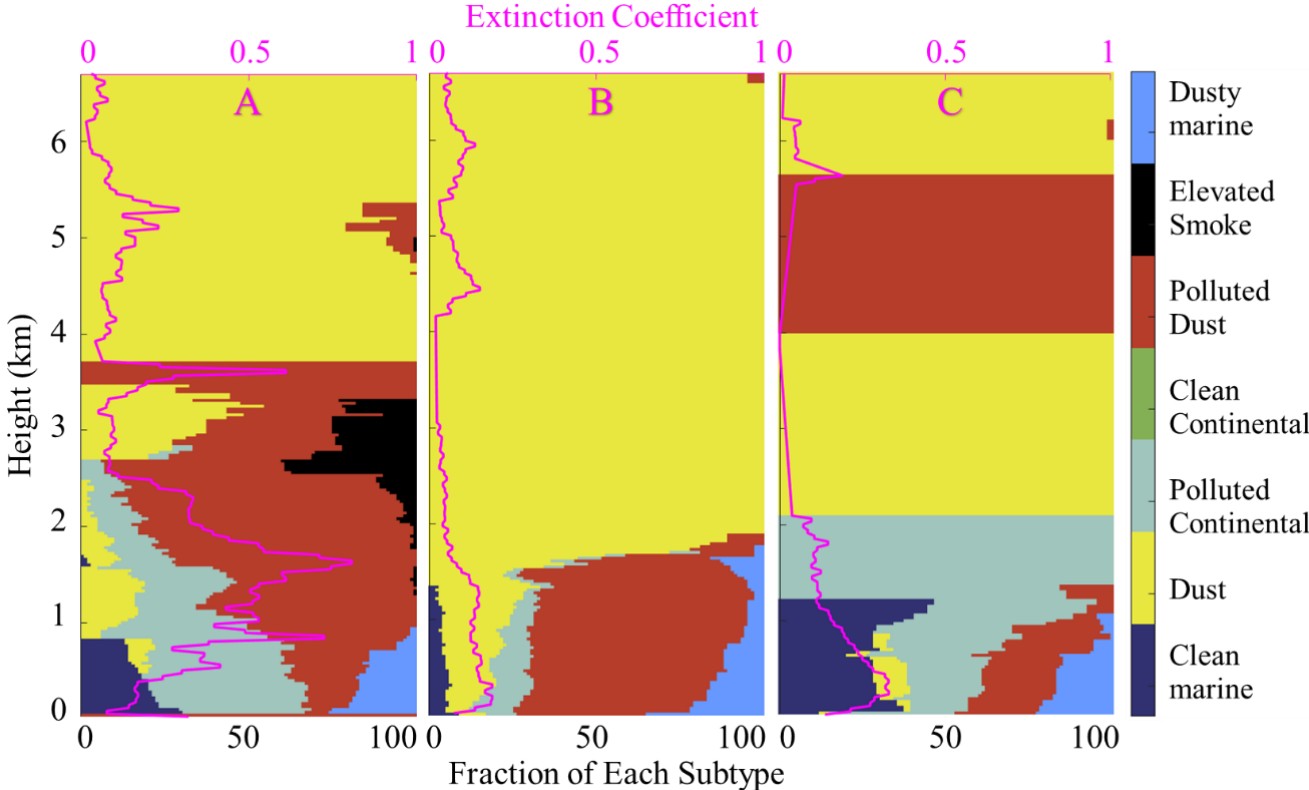

**Figure 6** Vertical distributions of each aerosol subtypes in region A, B, and C (see Fig. 5). Pink lines show profiles of average aerosol extinction coefficient (km$^{-1}$). The fraction is calculated as the occurrence of each subtype divided by the total occurrence of aerosols in each region.

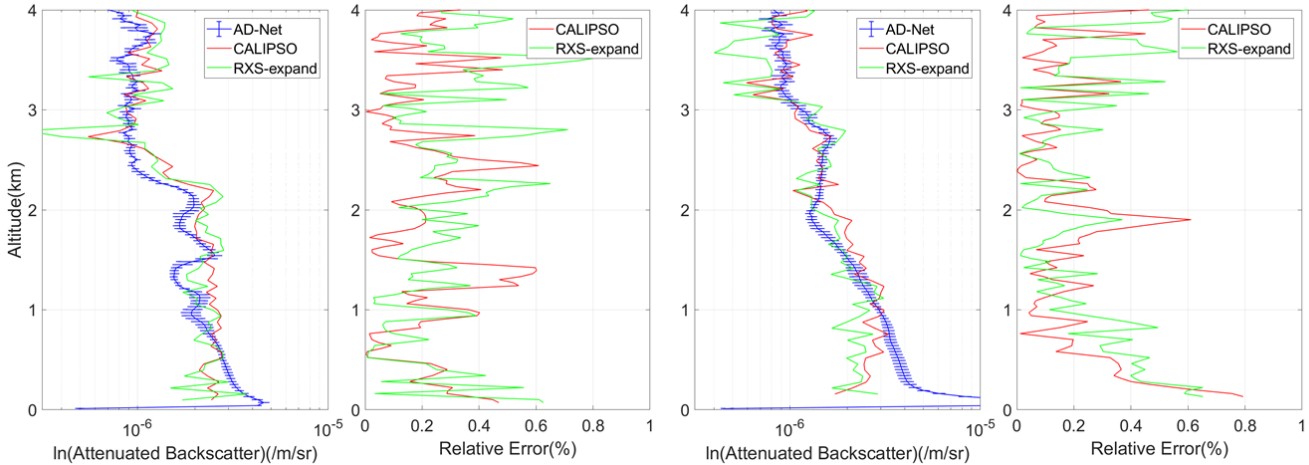

**Figure 7** Comparisons among ground-based lidar profiles of 532 nm attenuated backscatter coefficient products (units: m⁻¹sr⁻¹, averaged 2 hours within satellite overpass), CALIPSO profiles at shortest distance and RXS-expand profiles averaged 25 km around the location of Seoul station. The two plots on the left are from 7 March 2015, and the two plots on the right are from 24 April 2015.

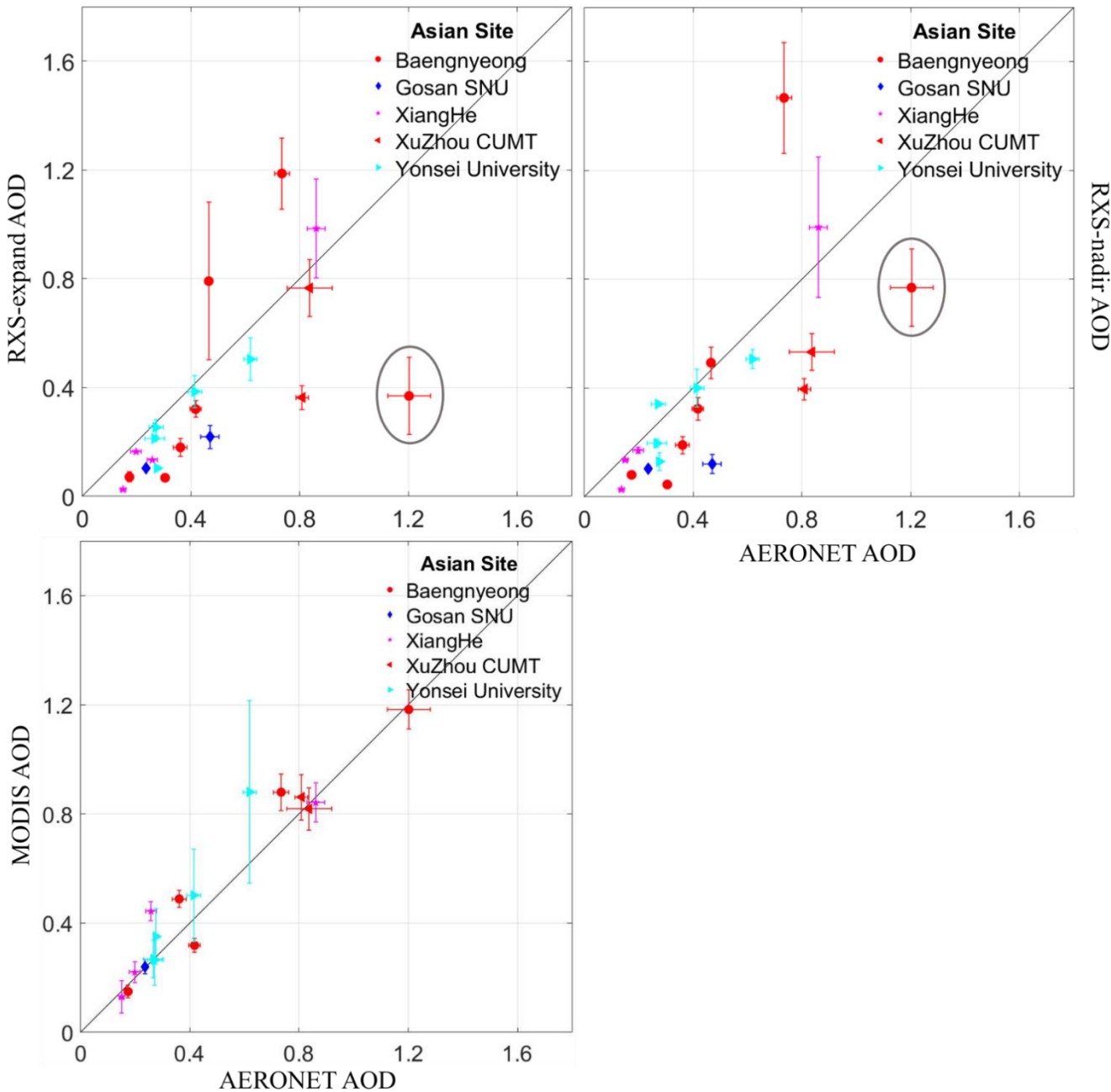

**Figure 8** Comparison of collocated measurements of RXS-expand and AERONET (upper left), RXS-nadir and AERONET (upper right), and MODIS and AERONET (lower left). Error bars in y-direction indicate standard deviations of RXS-expand, RXS-nadir or MODIS measurements within 50 km of the site, and in the x-direction AERONET measurements within $\pm 2$ hour of CALIPSO's passing. Grey circles mark a notable outlier.