# Peer review of "Analysis of Global Three-Dimensional Aerosol Structure with Spectral Radiance Matching"

_Atmospheric Measurement Techniques, 2019_

## Referee Comment (RC1) · Anonymous Referee #2 · 17 Jul 2019

The authors use the cloud construction algorithm developed by Barker et al. to construct vertical profile of aerosols. The algorithm seems to be exactly the same as the algorithm of Barker et al. with a different spatial resolution (a 5 km resolution for the lidar instrument). The study is focused on aerosol vertical profiles. They construct a 201 km wide of global aerosol profiles over two time periods. They test the algorithm by the same method discussed by Barker et al. They also evaluate the aerosol optical depth computed over nadir and off-nadir view regions and compare with the aerosol optical depth derived from AERONET.

I have serious concerns on this version. First, the algorithm developed by Barker et al. uses near infrared and infrared channels. I expect that the algorithm does not work for aerosols for two reasons. Aerosol signals in near IR and IR channels are very weak

and I am surprised that the algorithm works for aerosols. In addition, aerosol signal in the upward radiance is very small over land. The variability of visible radiance over land is dominated by the variability of surface reflectance. Again, I expect that the Barker algorithm does not work for aerosols especially over land. These lead me to wonder why the algorithm works. There is no physical explanation provided in the manuscript. The authors need to explain reasons why the authors expect that the Barker algorithm works for aerosols. The current manuscript is written in a way that the authors took the algorithm and found it works for aerosols.

Second, the authors compare the aerosol optical depth for validation of, essentially, the Barker's algorithm. Figure 7 clearly shows that MODIS AODs agree with AERONET AOD better than "nadir" and "expanded". MODIS instruments have a full coverage of Earth. Therefore, comparison of AOD presented in the paper does not show any advantage of the aerosol-constructing algorithm compared to MODIS. The authors need to show the validation of constructed aerosol vertical profiles, perhaps compared with ground-based observations to demonstrate the advantage of constructing aerosol profiles. The authors also need to show how the error in the vertical aerosol profiles reduces, compared with nadir view only, when constructed aerosol profiles are used.

---

## Author Comment (AC1) · 26 Jul 2019

We thank the reviewer for the comment. The algorithm we used, developed by Barker et al., utilized 0.62–0.67, 2.105–2.155, 8.4–8.7, and 11.77–12.27 $\mu$m channels from MODIS, or simply denoted as bands 1,7,29 and 32. Among these bands, bands 1 and 7 are also used for MODIS aerosol retrieval. Therefore, we believe the algorithm can also work for aerosols, as assumed in Barker et al. 2012. We did notice that the choice of channels might not be as beneficial for aerosols as they are for clouds—the sensitivity to aerosols might rely more on using visible channels, and the signal from aerosols could be much weaker. We had done following test in order to test and possibly optimize the algorithm for application on aerosols. We collected 30 days of CALIPSO profiles in 2015 with clear-sky condition and heavy aerosol loading at the east coast

[Figure]

of China. We expected the manually selected cloudless dataset with heavy loading events would give a clear answer to whether the algorithm could work for aerosols or not. We test the following combination of radiance bands: 1) using bands 1,7,29 and 32 used by Barker et al.; 2) using bands 1 and 7 only; 3) using visible bands 1, 2, 3 and 4. We tested the performance of using these combinations by reconstructing the profile with dead-zone setting for 30 and 100 km. A typical result is shown in the following figure, where the panel a is the original profile, panel b to d corresponds to combination 1 to 3, panel e shows the results of choosing the closest pixels outside the dead-zone. The results we got from this test indicate that the original combination used by Barker et al. could get a pretty successful reconstruction (on average 81.9% and 75.2% matching rate at 30 and 100 km, respectively), which means it can be used to construct aerosol vertical structure. In comparison, using visible channels only have lower matching rate (around 60-70%), especially when aloft aerosol layer are present. The closest pixel method, on the other hand, has very high matching rate at 30 km, but as the dead-zone range increases or if the aerosol layer is not continuous, the simple horizontal shift leads to more errors. In conclusion, we decided to use the channels selected in Barker et al. For reviewer's second concern, we want to clarify that we did not intend to get a 'better' quantification of AOD by expanding active profiles to nearby regions. After all, the active pixel being matched to the passive column could only provide an estimate of the column condition, which is not expected to be better than actual measurements MODIS made there. The advantage of this algorithm is really to help infer a profile and related vertical information, as shown in Figure 2 and Figure 6, which passive-only cannot obtain. As for reviewer's suggestion to validate the constructed aerosol vertical profiles with ground-based observations, we are sorry that our attempt to do so was not carried out, because there were no available lidar stations in the area. Therefore, we took one step back and compared the column total with ground-based AERONET sites. We did show in Figure 7 that constructed aerosol profiles made closer agreement with AERONET AOD than the nadir view only.

[Figure]

**Fig. 1.** Reconstruction of CALIPSO profile passing the east coast of China on January 3, 2015 with dead-zone setting for 100 km.

---

## Referee Comment (RC2) · Anonymous Referee #3 · 4 Sep 2019

Aerosol vertical structure (AVS) determines much of the climate impact of aerosol in the atmosphere, but it is difficult to measure with the spatial and temporal coverage needed for many applications. This study uses the spectral radiance matching (SRM) method, which infers AVS from column measurements by matching them to similar measurements with collocated vertical profiles, to construct global AVS from MODIS and CALIOP data. The paper is well organized, and the methods section is especially clearly written.

Although collocated AERONET data is widely used to validate MODIS aerosol, its use in a case study for this manuscript does not seem sufficient to evaluate the AVS retrieval. This is undoubtedly difficult, because the scarcity of AVS data that makes this study so valuable also leaves little basis for comparison. However, airborne field cam-

paigns and ground-based lidar networks do measure vertical profiles of clouds and aerosol on a smaller scale, and given the global MODIS/CALIOP record to choose from, it should be possible to use some of these measurements to validate the satellite AVS.

Specific comments:

Page 2, lines 1-23. I'm surprised not to see any mention here of ground-based lidar networks, which are sometimes used in combination with CALIOP data; or of the shorter-lived NASA CATS lidar that was aboard the ISS.

Page 6, lines 18-21. These cloud cover rates seem very low. For passive sensors, 70% is a reasonable ballpark estimate for the fraction of the globe covered by clouds at any given time. Most such clouds would occupy only a small part of the vertical column (and as the paper states, almost never at high altitudes) but the numbers still seem difficult to reconcile. Have you calculated the global cloud cover from the column perspective, for comparison?

Page 6, Figure 2. This is fascinating. It would be interesting to see a more detailed discussion of the contrast between 0-2 km and 2-4 km, which appear to distinguish local aerosol from aerosol undergoing long-range transport.

Page 8, Figure 3. This plot is somewhat difficult to read. A different color scheme may make the drop in the matching rate at the ITCZ easier to spot, but I'm having trouble seeing any other patterns.

Technical comments:

Page 6, line 14. "Losing".

Page 8, line 19. "CALIOP profiles".

---

## Author Comment (AC2) · 23 Sep 2019

We thank the reviewer for the comments and suggestions. We realize the importance to add comparison between constructed AVS from satellites to ground-based lidar measurements. We have refined our analysis and made changes to address other questions.

We thank reviewer's understanding of the difficulty of evaluating the AVS retrieval due to scarcity of AVS measurements, and we made our best effort to find measurements we were able to compare with.

We found the Asian dust and aerosol lidar observation network (AD-Net) when we looked for ground-based lidar stations with freely downloadable data. AD-Net is a lidar network for continuous observations of vertical distributions of Asian dust and other aerosols in East Asia. The sites contribute to the WMO GAW Program, and form the East Asian component of the GAW Aerosol Lidar Observation Network (GALION). Although cooperative stations in China didn't provide data sharing, we found some data that we were able to compare at Seoul station (37.5N,127.0E).

Seoul station has a standard lidar system in AD-Net, which is a two-wavelength (1064 nm, 532 nm) polarization sensitive (532 nm) Mie-scattering lidar, plus a 532 nm Raman (Shimizu et al. 2004). Based on the ground track, the A-Train sensors made overpass near the station for a total of 6 days during our case study in spring 2015. However, 4 out of these 6 days were heavily cloudy. For the remaining 2 days, March 7[th] and April 24[th], the comparisons among ground-based lidar profiles, CALIPSO profiles at shortest distance and RXS-expand profiles averaged 25 km around the location of Seoul station are shown in the following figure.

[Figure]

Figure 1    Comparisons among ground-based lidar profiles of 532 nm attenuated backscatter coefficient products (units: m$^{-1}$sr$^{-1}$, averaged 2 hour within satellite overpass), CALIPSO profiles at shortest distance and RXS-expand profile averaged 25 km around the location of Seoul station. The two plots on the left are from March 7[th], 2015, and the two plots on the right are from April 24[th], 2015.

The CALIPSO measurements we used for comparisons are level 1.5 data product of attenuated backscatter profiles, which clouds, overcast, surface, subsurface, and totally attenuated samples have been removed before being averaged to a 20 km horizontal resolution. In this case, RXS-expand profiles are based on the same products. The ground-based measurements used for comparison are the 532 nm attenuated backscatter coefficient products,

averaged within 2 hr before and after the satellite overpass with 15 min time resolution.

For the aerosol layer 0-4km above the ground, the relative error between CALIPSO profiles and ground station profiles are on average 21.6% on March 7[th], and 18.7% on April 24[th]. The distances between station and ground track are 51.0 km on the first day, and 50.1 km on the second. Between RXS-expand profiles and ground station profiles, the average relative error is 27.9% and 23.4%, respectively.

The results from the comparisons agreed in general. First of all, there were considerable disagreement between CALIPSO measurements and ground-based lidar measurements; in most studies, the differences were found to be around 20% (Mamouri et al. 2009, Wu et al. 2011, Kim et al. 2008, Chiang et al. 2011). For example, Mamouri et al. (2009) compared CALIPSO attenuated backscatter coefficient profiles with a ground-based lidar in Athens, Greece, and they found the agreement on the order of −10±12% for cloud-free daytime measurements between 3 and 10 km, while the differences between 1 and 3 km were much larger (−34±34%). In addition, we want to clarify that we did not intend to get a precise quantification of aerosol profile through the scene construction method. After all, the active pixel being matched to the passive column is intended only to provide an estimate of the column's vertical structure. We did expect, to some extent, the estimation could be improved through calculations with constrains such as the column AOD measured by passive sensor at the exact location of the recipient pixel, which will need a lot more work in the future.

As for reviewer's more specific comments, we made following changes to the content in the context and figures.

Page 2, lines 1-23. I'm surprised not to see any mention here of ground-based lidar networks, which are sometimes used in combination with CALIOP data; or of the shorter-lived NASA CATS lidar that was aboard the ISS.

We thank the author for the suggestion and we plan to add the following lines to the content, as a separate paragraph after the first one on that page.

*"The development of Lidar technology helped provide these vital missing piece of information. Ground-based lidar systems have been stationed at various locations and also used in field campaigns to measure the vertical and horizontal distribution of aerosols (Welton et al. 2000, Welton et al. 2002, Badarinath et al. 2010). Ground-based lidars provide measurements on the fixed locations on timescale of minutes to hours, depending on the specific type of lidar used in the experiment. Limited by the stationary setting, ground-based lidars could not achieve true global coverage, nevertheless, network of ground-based lidars (e.g. MPL-NET, EARLINET, AD-NET) provide key insights to atmospheric study and are involved in validation of satellite sensors (Kovacs et al. 2004, Mamouri et al. 2009, Pappalardo et al. 2010)."*

Page 6, lines 18-21. These cloud cover rates seem very low. For passive sensors, 70% is a reasonable ballpark estimate for the fraction of the globe

covered by clouds at any given time. Most such clouds would occupy only a small part of the vertical column (and as the paper states, almost never at high altitudes) but the numbers still seem difficult to reconcile. Have you calculated the global cloud cover from the column perspective, for comparison?

We thank the author for the question and comment. We will make it more clear in the revision that the numbers in Table.2 refers to cloud occurrence as percentage they occupied in the vertical column. We calculated this occurrence rate according to CALIPSO VFM products, which was scaled for vertical and horizontal resolution (Hunt et al. 2009). However, it is true that the numbers in the table underestimated the amount of clouds in actual atmosphere. As we stated, CALIPSO's signal can be totally attenuated beneath clouds and possibly making cloud layers below showed up as "no signal". The horizontal cloud coverage between 60°N and 60°S for the tested periods in April and September 2015 are 68.7% and 71.3%, respectively.

Page 6, Figure 2. This is fascinating. It would be interesting to see a more detailed discussion of the contrast between 0-2 km and 2-4 km, which appear to distinguish local aerosol from aerosol undergoing long-range transport.

We thank the author for the suggestion. We add more discussion about the contrast between 0-2 km and 2-4 km into the last paragraph on page 6. The paragraph is modified as following:

*"Height-resolved global AOD maps (averaged for a 2°×2° lat/long grid) based on the two selected periods are shown in Fig.2. In the near-surface layer, 2 km above ground level (AGL), in April, relatively high aerosol loadings are found in the cross-Atlantic African dust transport, Saudi Arabia, and India. In September, dust dynamics are much weaker but much biomass burning is apparent in the Brazilian Amazon and Southern Africa. This seasonal trend of dust and smoke is more obvious in the layer 2-4 km AGL. Aerosol in this layer aloft are expected to be undergoing long-range transport. In April, the thickest dust layers are found slightly inland of the western coast of Africa, around 12.5°N, 5.5°E, and in the center of Saudi Arabia around 24.5°N, 42.5°E. The shift of AOD distribution between surface layer and layer above is logical, and indicates the movement of dust layers as the aerosol loadings are transported towards the oceans. In September, this contrast is harder to observe as the dust dynamic is weaker, but similar trends are found in the biomass burning regions. In addition, persistent high aerosol loadings in both 0-2 km and 2-4 km AGL are found in India and the east coast of China with mixed sources of natural aerosols and pollutants. The results could be affected by the local topography. Marine aerosols are confined largely to the near-surface layer, with some vertical transport in Southeast Asia in September due to the Asian monsoon. The observed pattern is mostly consistent with other studies in terms of global distribution and seasonal variations (Martins et al., 2018;Liu et al., 2012;Chen et al., 2018)."*

Page 8, Figure 3. This plot is somewhat difficult to read. A different color scheme may make the drop in the matching rate at the ITCZ easier to spot, but

I'm having trouble seeing any other patterns.

We agree with the reviewer that Figure.3 is rather vague for the information we tried to convey. In fact, we think Figure.3 does not give enough extra information on its own, other than these results given in Figure.4. Therefore, we make a decision to remove this figure.

Technical comments:

Page 6, line 14. "Losing".

Page 8, line 19. "CALIOP profiles".

We thank the author for the comment, and these lines will be changed as suggested.

*"From 10 - 24 April and 14 - 29 September, CALIPSO functioned normally except during a boresight diagnostic and alignment on 18 September, losing about half of that day's data."*

*"With the complete datasets of CALIPO profiles, MODIS radiances and geolocation fields, construction based on the SRM method can be applied worldwide."*

Badarinath, K. V. S., S. K. Kharol, D. G. Kaskaoutis, A. R. Sharma, V. Ramaswamy & H. D. Kambezidis (2010) Long-range transport of dust aerosols over the Arabian Sea and Indian region A case study using satellite data and ground-based measurements. *Global and Planetary Change,* 72**,** 164-181.

Chiang, C.-W., S. K. Das, Y.-F. Shih, H.-S. Liao & J.-B. Nee (2011) Comparison of CALIPSO and ground-based lidar profiles over Chung-Li, Taiwan. *Journal of Quantitative Spectroscopy & Radiative Transfer,* 112**,** 197-203.

Hunt, W. H., D. M. Winker, M. A. Vaughan, K. A. Powell, P. L. Lucker & C. Weimer (2009) CALIPSO Lidar Description and Performance Assessment. *Journal of Atmospheric and Oceanic Technology,* 26**,** 1214-1228.

Kim, S. W., S. Berthier, J. C. Raut, P. Chazette, F. Dulac & S. C. Yoon (2008) Validation of aerosol and cloud layer structures from the space-borne lidar CALIOP using a ground-based lidar in Seoul, Korea. *Atmospheric Chemistry and Physics,* 8**,** 3705-3720.

Kovacs, T. A., M. P. McCormick, C. R. Trepte, D. M. Winker, A. Garnier & J. Pelon. 2004. Coordination of quid pro quo ground-based measurements of cloud and aerosol optical properties for validation of the CALIPSO mission. In *Conference on Lidar Remote Sensing for Industry and Environmental Monitoring V,* 281-289. Honolulu, HI.

Mamouri, R. E., V. Amiridis, A. Papayannis, E. Giannakaki, G. Tsaknakis & D. S. Balis (2009) Validation of CALIPSO space-borne-derived attenuated backscatter coefficient profiles using a ground-based lidar in Athens, Greece. *Atmospheric Measurement Techniques,* 2**,** 513-522.

Pappalardo, G., U. Wandinger, L. Mona, A. Hiebsch, I. Mattis, A. Amodeo, A. Ansmann, P. Seifert, H. Linne, A. Apituley, L. Alados Arboledas, D. Balis, A. Chaikovsky, G. D'Amico, F. De Tomasi, V. Freudenthaler, E. Giannakaki, A. Giunta, I. Grigorov, M. Iarlori, F. Madonna, R. E. Mamouri, L. Nasti, A. Papayannis, A. Pietruczuk, M. Pujadas, V. Rizi, F. Rocadenbosch, F. Russo, F. Schnell, N. Spinelli, X. Wang & M. Wiegner (2010) EARLINET correlative measurements for CALIPSO: First intercomparison results. *Journal of Geophysical*

*Research-Atmospheres,* 115.

Shimizu, A., N. Sugimoto, I. Matsui, K. Arao, I. Uno, T. Murayama, N. Kagawa, K. Aoki, A. Uchiyama & A. Yamazaki (2004) Continuous observations of Asian dust and other aerosols by polarization lidars in China and Japan during ACE-Asia. *Journal of Geophysical Research-Atmospheres,* 109.

Welton, E. J., K. J. Voss, H. R. Gordon, H. Maring, A. Smirnov, B. Holben, B. Schmid, J. M. Livingston, P. B. Russell, P. A. Durkee, P. Formenti & M. O. Andreae (2000) Ground-based lidar measurements of aerosols during ACE-2: instrument description, results, and comparisons with other ground-based and airborne measurements. *Tellus Series B-Chemical and Physical Meteorology,* 52, 636-651.

Welton, E. J., K. J. Voss, P. K. Quinn, P. J. Flatau, K. Markowicz, J. R. Campbell, J. D. Spinhirne, H. R. Gordon & J. E. Johnson (2002) Measurements of aerosol vertical profiles and optical properties during INDOEX 1999 using micropulse lidars. *Journal of Geophysical Research-Atmospheres,* 107.

Wu, D., Z. Wang, B. Wang, J. Zhou & Y. Wang (2011) CALIPSO validation using ground-based lidar in Hefei (31.9A degrees N, 117.2A degrees E), China. *Applied Physics B-Lasers and Optics,* 102, 185-195.

---

## Referee Comment (RC3) · Anonymous Referee #1 · 26 Sep 2019

The authors present a method that expands the aerosol vertical profiles retrieved with nadir-pointing lidars to locations away from the lidar's nadir track that is based on matching radiances with a colocated imaging multispectral radiometer. The method, which includes a self contained ability to test the reliability of the profiles constructed off the track, is tested using CALIPSO and MODIS data. Finally, a case study is performed, that presents scientific results.

Overall, I believe the manuscript is generally well written and contributes a useful and potentially important scientific tool although there are several minor, mostly grammatical errors, that I have listed. I therefore, recommend that it be considered for publication after minor revisions.

[Figure]

**Issues throughout the paper:**

- In some cases there is a space between a value and the unit km and in some cases there is no space. Using a space is preferred but in any case it should be consistent.

- Use of LaTex '-' vs '–' (two minus signs) is not used properly in some places.

- Equations before 'where' should have commas after them.

- There are cases of Table.X and Figure.X which obviously should not have a period. There are also cases of Fig.X and Tab.X that don't have a space after the period which they should have.

- In table captions the period is missing after the table number.

- Maybe use 'lon' instead of 'long' to be consistent with the 3 letter use of 'lat'.

- In almost all cases of parenthetical cites with more than one cite there is no space after the semicolon. There should be.

- In the table titles, except for the 'Table X.' the rest of the caption should not be bold.

- Please indicate the processing versions of the CALIPSO and MODIS products.

**Comments with a particular location:**

- **Page 1, Line 28:** The cite uses [] when it should use ().

- **Page 2, Line 5:** The reference used here is rather old and is cloud specific. What about Levy et al. (2013).

- **Page 4, Line 5:** The MODIS radiances are also provided at 250 m (bands 1–2) and 500 m bands (1–7) and I am pretty sure these are used for the official aerosol product.

- **Page 4, Line 6:** Again the the references here could be better and more up-to-date. Consider using Levy et al. (2013) and Platnick et al. (2017).

- **Page 4, Line 26:** The operators and brackets should not be italicised.

- **Page 4, Eq 1:** Use italics (math mode) for variables F(i,j;m) → F$(i, j; m)$.

- **Page 4, Eq 1:** In the summation notation the lower variable is an index. The upper variable should be the upper value of the index. In this case both are the same variable.

- **Page 4, Line 30:** $K = 4$? Related to previous comment.

- **Page 4, Line 31:** I think I understand why these bands are chosen but maybe have a bit of explanation.

- **Page 5, Eq 2:** Use italics (math mode) for variables D(i,j;m) → D$(i, j; m)$.

- **Page 5, Line 9:** The asterisk in $(m^*, 0)$ is weird.

- **Page 5, Line 12:** 'of construction' → 'of the construction'.

- **Page 6, Eq 5:** Inconsistent use of italics. MR should not be italicized (use \mathrm) and $N$ should be italicized. Note other usages of MR below.

- **Page 6, Line 30:** 'km AGL' → 'km above AGL'.

- **Page 7, Line 15:** 'donors meets' → 'donors that meet'.

- **Page 7, Line 18:** 'air column' → 'air columns'.

- **Page 7, Line 19:** 'air column' → 'air columns'.

- **Page 7, Line 29:** 'This is resulted' → 'This results'.

- **Page 7, Line 29:** 'of TBM' → 'of the TBM'.

- **Page 7, Line 32:** 'by total' → 'by the total'.

- **Page 7, Line 33:** 'in reconstructed' → 'in the reconstructed'.

- **Page 8, Line 1:** 'in reconstructed' → 'in the reconstructed'.

- **Page 8, Line 11:** Use LaTex \times instead of 'x'.

- **Page 8, Line 16:** 'In reality, this' → 'This'.

- **Page 8, Line 29:** Use LaTex \times instead of 'x'.

- **Page 8, Line 30:** 'surrounding Bohai' → 'surrounding the Bohai'.

- **Page 9, Line 13:** 'heating. Average' → 'heating. The average'.

- **Page 9, Line 16:** 'C, average' → 'C, the average'.

- **Page 9, Line 20:** 'AOD, especially' → 'AOD, with especially'.

- **Page 9, Line 26:** 'with CALIPSO' → 'with the CALIPSO'.

- **Page 9, Line 27:** 'hour of' → 'hours of'.

- **Page 9, Line 27:** 'using 10' → 'using the 10'.

- **Page 9, Line 31:** 'have larger' → 'have a larger'.

- **Page 10, Line 9:** 'to alter our' → 'to improve our'.

- **Page 10, Line 10:** 'for study' → 'for the study'.

- **Page 10, Line 12:** 'using SRM' → 'using the SRM'.

- **Page 10, Line 16:** '6.%' → '6%'.

- **Page 10, Line 18:** 'with sufficient' → 'with a sufficient'.

- **Page 10, Line 25:** 'on SRM' → 'on the SRM'.

- **Page 10, Line 25:** 'a power' → 'an important'.

- **Page 10, Line 26:** 'well off' → 'not'.

- **Page 16, Table 3:** Identify the abbreviations in the headers.

- **Page 20, Figure 3:** 'with 30km' → 'with a 30km'.

- **Page 20, Figure 3:** \sigma should be in italicized.

**References**

R. C. Levy, Mattoo, L. A. Munchak, L. A. Remer, A. M. Sayer, and N. C. Hsu. The collection 6 modis aerosol products over land and ocean. *Atmospheric Measurement Techniques*, 6: 159–259, 2013. doi: 10.5194/amtd-6-159-2013.

Steven Platnick, Kerry G. Meyer, Michael D. King, Galina Wind, Nandana Amarasinghe, Benjamin Marchant, G. Thomas Arnold, Zhibo Zhang, Paul A. Hubanks, Robert E. Holz, Ping Yang, William L. Ridgway, and Jérôme Riedi. The MODIS cloud optical and microphysical products: Collection 6 updates and examples from Terra and Aqua. *IEEE Transactions on Geoscience and Remote Sensing*, 55(1):502–525, January 2017. doi: 10.1109/TGRS.2016.2610522.

---

## Author Comment (AC3) · 27 Sep 2019

We thank the reviewer again for the comments and suggestions. We apologize for the first version of response being kind of vague on the changes we made. Here we include a more specific list of changes in manuscript.

To address the reviewer's comment on comparisons with ground-based lidars, the following experiment and discussion has been added to Section 4.3, Page 10-11.

"In the three-month period, the Asian dust and aerosol lidar observation network (AD-Net) site at Seoul, Korea (37.5N,127.0E) provided measurements of atmospheric profiles that we were able to compare with RXS-expand. Seoul station has a standard lidar system in AD-Net, which is a two-wavelength (1064 nm, 532 nm) polarization sensitive

(532 nm) Mie-scattering lidar, plus a 532 nm Raman (Shimizu et al., 2004). Based on the ground track, the A-Train sensors made overpass near the station for a total of 6 days during spring 2015. However, 4 out of these 6 days were heavily cloudy. For the remaining 2 days, March 7th and April 24th, the comparisons among ground-based lidar profiles, CALIPSO profiles at shortest distance and RXS-expand profiles averaged 25 km around the location of Seoul station are shown Fig.6. The CALIPSO measurements used for comparisons are level 1.5 data products of attenuated backscatter profiles, which clouds, overcast, surface, subsurface, and totally attenuated samples have been removed before being averaged to a 20 km horizontal resolution. In this case, RXS-expand profiles are based on the same products. The ground-based measurements used for comparison are the 532 nm attenuated aerosol backscatter coefficient products, averaged within 2 hr before and after the satellite overpass with 15 min time resolution.

For the aerosol layer 0-4 km above the ground, the relative error between CALIPSO profiles and ground station profiles are on average 21.6% on March 7th, and 18.7% on April 24th. The distances between station and ground track are 51.0 km on the first day, and 50.1 km on the second. Between RXS-expand profiles and ground station profiles, the average relative error is 27.9% and 23.4%, respectively. The results from the comparisons agreed in general. Previous studies found that there were considerable disagreement between CALIPSO measurements and ground-based lidar measurements; in most studies, the differences were found to be around 20% (Mamouri et al., 2009;Wu et al., 2011;Kim et al., 2008;Chiang et al., 2011). For example, Mamouri et al. (2009) compared CALIPSO attenuated backscatter coefficient profiles with a ground-based lidar in Athens, Greece, and they found the agreement on the order of $-10\pm12\%$ for cloud-free daytime measurements between 3 and 10 km, while the differences between 1 and 3 km were much larger ($-34\pm34\%$). In addition, the scene construction method in this work is not intended to get a precise quantification of aerosol profile, but to provide an estimate of the column's vertical structure. We did expect, to some extent, the estimation could be improved through calculations with constrains such as the column

AOD measured by passive sensor at the exact location of the recipient pixel, which will need a lot more work in the future."

Please see Figure.6 in the marked-up manuscript. The complete caption is following. Figure 6 Comparisons among ground-based lidar profiles of 532 nm attenuated backscatter coefficient products (units: m-1sr-1, averaged 2 hour within satellite overpass), CALIPSO profiles at shortest distance and RXS-expand profiles averaged 25 km around the location of Seoul station. The two plots on the left are from March 7th, 2015, and the two plots on the right are from April 24th, 2015.

Specific comments:

Page 2, lines 1-23. I'm surprised not to see any mention here of ground-based lidar networks, which are sometimes used in combination with CALIOP data; or of the shorter-lived NASA CATS lidar that was aboard the ISS.

We thank the author for the suggestion and we plan to add the following lines to the content, as a separate paragraph after the first one on page 2, lines 10-16. "The development of Lidar technology helped provide these vital missing piece of information. Ground-based lidar systems have been stationed at various locations and also used in field campaigns to measure the vertical and horizontal distribution of aerosols (Welton et al., 2000;Welton et al., 2002;Badarinath et al., 2010). Ground-based lidars provide measurements on the fixed locations on timescale of minutes to hours, depending on the specific type of lidar used in the experiment. Limited by the stationary setting, ground-based lidars could not achieve true global coverage, nevertheless, network of ground-based lidars (e.g. MPL-NET, EARLINET, AD-NET) provide key insights to atmospheric study and are involved in validation of satellite sensors (Kovacs et al., 2004;Mamouri et al., 2009;Pappalardo et al., 2010)."

Page 6, lines 18-21. These cloud cover rates seem very low. For passive sensors, 70% is a reasonable ballpark estimate for the fraction of the globe covered by clouds at any given time. Most such clouds would occupy only a small part of the vertical

column (and as the paper states, almost never at high altitudes) but the numbers still seem difficult to reconcile. Have you calculated the global cloud cover from the column perspective, for comparison?

We thank the author for the question and comment. We made changes to paragraph on page 6-7 to make it more clear that the numbers in Table.2 refers to cloud occurrence as percentage they occupied in the vertical column, and also addressing reviewer's other questions.

"Table.2 summarizes frequencies of occurrences of atmosphere conditions. These numbers refer to the occurrence of atmospheric features as percentage they occupied in the vertical column. We calculated this occurrence rate according to CALIPSO VFM products, which was then scaled for vertical and horizontal resolution of the products (Hunt et al., 2009).The majority of conditions were identified as clear. Above 8.2 km clear arrays occur over 90% of the time. Aerosols and clouds occurred below 8.2 km, in about 7% and 5% of the cells, respectively. Note that arrays identified as 'no signal' represent $16 - 18\%$ of cells in this layer; CALIPSO's signal can be totally attenuated beneath opaque clouds and certain aerosols. This indicates that the numbers in the table likely underestimated the amount of clouds in actual atmosphere. After removing elements identified as 'no signal', surface, and sub-surface, aerosols and clouds occupied $4.43 - 4.52\%$ and $5.35 - 6.15\%$ of the cells; clear-skies account for the remainder. The horizontal cloud coverage between $60°N$ and $60°S$ for the tested periods in April and September 2015 are 68.7% and 71.3%, respectively."

Page 6, Figure 2. This is fascinating. It would be interesting to see a more detailed discussion of the contrast between 0-2 km and 2-4 km, which appear to distinguish local aerosol from aerosol undergoing long-range transport.

We thank the author for the suggestion. We add more discussion about the contrast between 0-2 km and 2-4 km into the last paragraph on page 7, lines 10-23. The paragraph is modified as following: "Height-resolved global AOD maps (averaged for

a 2°×2° lat/long grid) based on the two selected periods are shown in Fig.2. In the near-surface layer, 2 km above ground level (AGL), in April, relatively high aerosol loadings are found in the cross-Atlantic African dust transport, Saudi Arabia, and India. In September, dust dynamics are much weaker but much biomass burning is apparent in the Brazilian Amazon and Southern Africa. This seasonal trend of dust and smoke is more obvious in the layer 2-4 km AGL. Aerosol in this layer aloft are expected to be undergoing long-range transport. In April, the thickest dust layers are found slightly inland of the western coast of Africa, around 12.5°N, 5.5°E, and in the center of Saudi Arabia around 24.5°N, 42.5°E. The shift of AOD distribution between surface layer and layer above is logical, and indicates the movement of dust layers as the aerosol loadings are transported towards the oceans. In September, this contrast is harder to observe as the dust dynamic is weaker, but similar trends are found in the biomass burning regions. In addition, persistent high aerosol loadings in both 0-2 km and 2-4 km AGL are found in India and the east coast of China with mixed sources of natural aerosols and pollutants. The results could be affected by the local topography. Marine aerosols are confined largely to the near-surface layer, with some vertical transport in Southeast Asia in September due to the Asian monsoon. The observed pattern is mostly consistent with other studies in terms of global distribution and seasonal variations (Martins et al., 2018;Liu et al., 2012;Chen et al., 2018)."

Page 8, Figure 3. This plot is somewhat difficult to read. A different color scheme may make the drop in the matching rate at the ITCZ easier to spot, but I'm having trouble seeing any other patterns. We agree with the reviewer that Figure.3 is rather vague for the information we tried to convey. In fact, we think Figure.3 does not give enough extra information on its own, other than these results given in Figure.4. Therefore, we make a decision to remove this figure and change the order of figures accordingly.

Page 6, line 14. "Losing".

Page 8, line 19. "CALIOP profiles".

We thank the author for the comment, and these lines will be changed as suggested.

"From 10 - 24 April and 14 - 29 September, CALIPSO functioned normally except during a boresight diagnostic and alignment on 18 September, losing about half of that day's data."

"With the complete datasets of CALIOP profiles, MODIS radiances and geolocation fields, construction based on the SRM method can be applied worldwide."

Please also note the supplement to this comment:
https://www.atmos-meas-tech-discuss.net/amt-2019-182/amt-2019-182-AC3-supplement.pdf

―――――――――――――――――――――――

[revised manuscript text omitted]

---

## Author Comment (AC4) · 27 Sep 2019

We thank the reviewer for the time spent to evaluate our work and we thank him/her for the very useful and detailed comments.

We made following changes in the revision according to reviewer's comment.

For the issues throughout paper reviewer pointed out:

–We have been consistent to use a space between a value and the unit.

–We are not using LaTex, but we have replaced all '–' in the context, and been consistent to use '-'.

–We have put commas after Equations before 'where'.

[Figure]

–We have fixed the format of using abbreviation or not abbreviated form of 'Figure' and 'Table' in context.

–We have fixed the format of using abbreviation or not abbreviated form of 'Figure' and 'Table' in context.

–We have added period after the table number in table captions.

–We thank author's recommendation, but we think both 'long' and 'lon' are commonly used for abbreviation of longitude.

–The citation in the context is done through Cite While You Write™. We thank the reviewer for pointing it out and have added space in between cited works.

–We have made sure that only 'Table X.' is bold in the title.

–We have added to the data part that all CALIPSO products used in the work are from Version 4.20, and MODIS products used in the work are from MODIS Collection 6

Specific Comment

–Page 1, Line 28: The cite uses [] when it should use ().

The brackets have been changed.

–Page 2, Line 5: The reference used here is rather old and is cloud specific. What about Levy et al. (2013).

The suggested reference has been added.

–Page 4, Line 5: The MODIS radiances are also provided at 250 m (bands 1–2) and 500 m bands (1–7) and I am pretty sure these are used for the official aerosol product.

We thank reviewer for the comment. MODIS visible bands are provided with better resolutions, however, 1 km resolution is used to be consistent with infrared bands.

–Page 4, Line 6: Again the references here could be better and more up-to-date. Consider using Levy et al. (2013) and Platnick et al. (2017).

The suggested references have been added.

–Page 4, Line 26: The operators and brackets should not be italicized.

The operators and brackets have been changed.

–Page 4, Eq 1: Use italics (math mode) for variables F(i,j;m) → $F(i,j;m)$.

The equation has been changed.

–Page 4, Eq 1: In the summation notation the lower variable is an index. The upper variable should be the upper value of the index. In this case both are the same variable.

We have changed the upper value of the index to 4.

–Page 4, Line 30: K = 4? Related to previous comment.

Same as the last question, we have changed the upper value of the index to 4.

–Page 4, Line 31: I think I understand why these bands are chosen but maybe have a bit of explanation.

We thank the review for the comment. We have added following explanation to the context:

"The bands are chosen for their widely accepted usage in retrieving cloud properties, including cloud cover, cloud top properties (CTP/CTT/CTH), and cloud phase (Ackerman et al. 1998, Baum et al. 2012, Baum et al. 2000) and aerosol properties (Sayer et al. 2014, Levy et al. 2013, Remer et al. 2013)."

–Page 5, Eq 2: Use italics (math mode) for variables D(i,j;m) → $D(i,j;m)$.

The equation has been changed.

–Page 5, Line 9: The asterisk in (m âĹŮ ,0) is weird.

We thank the reviewer for the comment. The asterisk is used to specify that from all potential donors, the most suitable match is chosen to be the donor for the specific recipient. We add lines to clarify this in the context.

–Page 5, Line 12: 'of construction' → 'of the construction'.

The line has been changed accordingly.

Page 6, Eq 5: Inconsistent use of italics. MR should not be italicized (use\mathrm) and N should be italicized. Note other usages of MR below.

The equation has been changed, as well as the usage in the following context.

–Page 6, Line 30: 'km AGL' → 'km above AGL'.

Since the AGL stands for above ground level, we think no change is needed here.

–Page 7, Line 15: 'donors meets' → 'donors that meet'.

The line has been changed accordingly.

–Page 7, Line 18: 'air column' → 'air columns'.

The line has been changed accordingly.

–Page 7, Line 19: 'air column' → 'air columns'.

The line has been changed accordingly.

–Page 7, Line 29: 'This is resulted' → 'This results'.

The line has been changed accordingly.

–Page 7, Line 29: 'of TBM' → 'of the TBM'.

The line has been changed accordingly.

–Page 7, Line 32: 'by total' → 'by the total'.

The line has been changed accordingly.

–Page 7, Line 33: 'in reconstructed' → 'in the reconstructed'.

The line has been changed accordingly.

–Page 8, Line 1: 'in reconstructed' → 'in the reconstructed'.

The line has been changed accordingly.

–Page 8, Line 11: Use LaTex \times instead of 'x'.

The manuscript is not prepared in LaTex. We apologize for the inconvenience.

–Page 8, Line 16: 'In reality, this' → 'This'.

The line has been changed accordingly.

–Page 8, Line 29: Use LaTex \times instead of 'x'.

The manuscript is not prepared in LaTex. We apologize for the inconvenience.

–Page 8, Line 30: 'surrounding Bohai' → 'surrounding the Bohai'.

The line has been changed accordingly.

–Page 9, Line 13: 'heating. Average' → 'heating. The average'.

The line has been changed accordingly.

–Page 9, Line 16: 'C, average' → 'C, the average'.

The line has been changed accordingly.

–Page 9, Line 20: 'AOD, especially' → 'AOD, with especially'.

The line has been changed accordingly.

–Page 9, Line 26: 'with CALIPSO' → 'with the CALIPSO'.

The line has been changed accordingly.

–Page 9, Line 27: 'hour of' → 'hours of'.

The line has been changed accordingly.

–Page 9, Line 27: 'using 10' → 'using the 10'.

The line has been changed accordingly.

–Page 9, Line 31: 'have larger' → 'have a larger'.

The line has been changed accordingly.

–Page 10, Line 9: 'to alter our' → 'to improve our'.

The line has been changed accordingly.

–Page 10, Line 10: 'for study' → 'for the study'.

The line has been changed accordingly.

–Page 10, Line 12: 'using SRM' → 'using the SRM'.

The line has been changed accordingly.

–Page 10, Line 16: '6.%' → '6%'.

The line has been changed accordingly.

–Page 10, Line 18: 'with sufficient' → 'with a sufficient'.

The line has been changed accordingly.

–Page 10, Line 25: 'on SRM' → 'on the SRM'.

The line has been changed accordingly.

–Page 10, Line 25: 'a power' → 'an important'.

The line has been changed accordingly.

–Page 10, Line 26: 'well off' → 'not'. The line has been changed accordingly.

–Page 16, Table 3: Identify the abbreviations in the headers.

Thank for the suggestion, we add to the footnote of the table that 'No Sig' stands for 'no signal' and 'Surf' stands for 'surface and subsurface' portions of the measured columns.

–Page 20, Figure 3: 'with 30km' → 'with a 30km'.

The line has been changed accordingly.

–Page 20, Figure 3: \sigma should be in italicized.

Ackerman, S. A., K. I. Strabala, W. P. Menzel, R. A. Frey, C. C. Moeller & L. E. Gumley (1998) Discriminating clear sky from clouds with MODIS. Journal of Geophysical Research-Atmospheres, 103, 32141-32157.

Baum, B. A., W. P. Menzel, R. A. Frey, D. C. Tobin, R. E. Holz, S. A. Ackerman, A. K. Heidinger & P. Yang (2012) MODIS Cloud-Top Property Refinements for Collection 6. Journal of Applied Meteorology and Climatology, 51, 1145-1163.

Baum, B. A., P. F. Soulen, K. I. Strabala, M. D. King, S. A. Ackerman, W. P. Menzel & P. Yang (2000) Remote sensing of cloud properties using MODIS airborne simulator imagery during SUCCESS 2. Cloud thermodynamic phase. Journal of Geophysical Research-Atmospheres, 105, 11781-11792.

Levy, R. C., S. Mattoo, L. A. Munchak, L. A. Remer, A. M. Sayer, F. Patadia & N. C. Hsu (2013) The Collection 6 MODIS aerosol products over land and ocean. Atmospheric Measurement Techniques, 6, 2989-3034.

Remer, L. A., S. Mattoo, R. C. Levy & L. A. Munchak (2013) MODIS 3 km aerosol product: algorithm and global perspective. Atmospheric Measurement Techniques, 6, 1829-1844.

Sayer, A. M., L. A. Munchak, N. C. Hsu, R. C. Levy, C. Bettenhausen & M. J. Jeong (2014) MODIS Collection 6 aerosol products: Comparison between Aqua's e-Deep

Blue, Dark Target, and "merged" data sets, and usage recommendations. Journal of Geophysical Research-Atmospheres, 119, 13965-13989.

Please also note the supplement to this comment:
https://www.atmos-meas-tech-discuss.net/amt-2019-182/amt-2019-182-AC4-supplement.pdf

**Supplement:**

[revised manuscript text omitted]

---

## Author Comment (AC5) · 29 Sep 2019

Dear Reviewers and Editor,

We would like to thank all the reviewers again for the time spent to evaluate our work and for the useful and constructive comments they gave which help to improve our work. We also acknowledge the editor. Here we submit a final version of manuscript which includes all proposed changes during the discussion that we think are necessary to be made in the revised manuscript. Some final changes are made in related abstract/conclusions, regarding the additional comparison with ground-based lidar measurements. Thanks again for your contribution.

[Figure]

Please also note the supplement to this comment:
https://www.atmos-meas-tech-discuss.net/amt-2019-182/amt-2019-182-AC5-
supplement.pdf
* * *
[Figure]

**Supplement:**

[revised manuscript text omitted]

---

## Author Response (AR1)

Dear Reviewers and Editor,

We would like to thank all the reviewers again for the time spent to evaluate our work and for the useful and constructive comments they gave which help to improve our work. We also acknowledge the editor. Here we submit a final version of manuscript which includes all proposed changes during the discussion that we think are necessary to be made in the revised manuscript. Thanks again for your contribution.

Note that our answers are in blue in the following text. The actual changes made in the manuscript is in *italics*.

Anonymous Referee #1

The authors present a method that expands the aerosol vertical profiles retrieved with nadir-pointing lidars to locations away from the lidar's nadir track that is based on matching radiances with a collocated imaging multispectral radiometer. The method, which includes a self-contained ability to test the reliability of the profiles constructed off the track, is tested using CALIPSO and MODIS data. Finally, a case study is performed, that presents scientific results.

Overall, I believe the manuscript is generally well written and contributes a useful and potentially important scientific tool although there are several minor, mostly grammatical errors, that I have listed. I therefore, recommend that it be considered for publication after minor revisions.

Issues throughout the paper:

● In some cases, there is a space between a value and the unit km and in some cases there is no space. Using a space is preferred but in any case it should be consistent.

Thanks to this comment, we have made changes throughout the paper to be consistent to use a space between a value and the unit.

● Use of LaTex '-' vs '–' (two minus signs) is not used properly in some places.

We are not using LaTex, but we have replaced all '–' in the context, and been consistent to use '-'.

● Equations before 'where' should have commas after them.

We have put commas after Equations 1-5, since they are all connected with a 'where' or 'which'.

● There are cases of Table.X and Figure.X which obviously should not have a period. There are also cases of Fig.X and Tab.X that don't have a space after the period which they should have.

We have fixed the format of using abbreviated or not abbreviated form of 'Figure' and 'Table' in context.

- In table captions the period is missing after the table number.

We have added period after the table number in table captions.

- Maybe use 'lon' instead of 'long' to be consistent with the 3 letter use of 'lat'.

5   We thank author's recommendation, but we think both 'long' and 'lon' are commonly used for abbreviation of longitude, and we decide not to change it.

- In almost all cases of parenthetical cites with more than one cite there is no space after the semicolon. There should be.

The citation in the context is done through Cite While You Write™. We thank the reviewer for pointing

10  it out and have added space in between cited works.

- In the table titles, except for the 'Table X.' the rest of the caption should not be bold.

We have made sure that only 'Table X.' is bold in the title.

- Please indicate the processing versions of the CALIPSO and MODIS products.

We have added to the data part that all CALIPSO products used in the work are from Version 4.20, and

15  MODIS products used in the work are from MODIS Collection 6. The following lines have been added to the text on page 4, lines 14-15:

*"CALIPSO products used in the work are from Version 4.20."*

The following lines have been added to the text on page 4, lines 19-20:

*"MODIS products used in the work are from MODIS Collection 6."*

Comments with a particular location:

- Page 1, Line 28: The cite uses [] when it should use ().

The brackets have been changed.

- Page 2, Line 5: The reference used here is rather old and is cloud specific. What about Levy et al.

25   (2013).

We thank reviewer for the suggestion. The suggested reference has been added to the citation.

*"R. C. Levy, Mattoo, L. A. Munchak, L. A. Remer, A. M. Sayer, and N. C. Hsu. The collection 6 modis aerosol products over land and ocean. Atmospheric Measurement Techniques, 6:159–259, 2013. doi: 10.5194/amtd-6-159-2013."*

- Page 4, Line 5: The MODIS radiances are also provided at 250 m (bands 1–2) and 500 m bands (1–7) and I am pretty sure these are used for the official aerosol product.

We thank reviewer for the comment. MODIS visible bands are provided with better resolutions, however, 1 km resolution is used in the work to be consistent with infrared bands.

- Page 4, Line 6: Again the the references here could be better and more up-to-date. Consider using Levy et al. (2013) and Platnick et al. (2017).

We thank reviewer for the suggestion. The suggested references have been added to the citation.

*"R. C. Levy, Mattoo, L. A. Munchak, L. A. Remer, A. M. Sayer, and N. C. Hsu. The collection 6 modis aerosol products over land and ocean. Atmospheric Measurement Techniques, 6: 159–259, 2013. doi: 10.5194/amtd-6-159-2013.*

*Steven Platnick, Kerry G. Meyer, Michael D. King, Galina Wind, Nandana Amarasinghe, Benjamin Marchant, G. Thomas Arnold, Zhibo Zhang, Paul A. Hubanks, Robert E. Holz, Ping Yang, William L. Ridgway, and Jérôme Riedi. The MODIS cloud optical and microphysical products: Collection 6 updates and examples from Terra and Aqua. IEEE Transactions on Geoscience and Remote Sensing, 55(1):502–525, January 2017. doi:10.1109/TGRS.2016.2610522."*

- Page 4, Line 26: The operators and brackets should not be italicised.

The operators and brackets have been changed.

- Page 4, Eq 1: Use italics (math mode) for variables F(i,j;m) → F(i,j;m).

The equation has been changed.

$$F(i, j; m) = \sum_{k=1}^{K=4} \left( \frac{r_k(i,j) - r_k(m,0)}{r_k(i,j)} \right)^2 ; m \in [i - m_1, i + m_2],$$ (1)

- Page 4, Eq 1: In the summation notation the lower variable is an index. The upper variable should be the upper value of the index. In this case both are the same variable.

We have changed the upper value of the index to 4. Please see the change above.

- Page 4, Line 30: K = 4? Related to previous comment.

We have changed the upper value of the index to 4. Please see the changes above.

- Page 4, Line 31: I think I understand why these bands are chosen but maybe have a bit of explanation.

We thank the review for the comment. We have added following explanation to the text on page 5, lines 14-16:

*"The bands are chosen for their widely accepted usage in retrieving cloud properties, including cloud cover, cloud top properties (CTP/CTT/CTH), and cloud phase (Ackerman et al., 1998;Baum et al., 2012;Baum et al., 2000) as well as aerosol properties (Sayer et al., 2014;Levy et al., 2013;Remer et al., 2013)."*

- Page 5, Eq 2: Use italics (math mode) for variables D(i,j;m) → D(i,j;m).

The equation has been changed.

$$m_1 = m_2 = \begin{cases} 200; & D_m \leq 30 \\ 200 + D_m; & D_m > 30 \end{cases}, \tag{2}$$

- Page 5, Line 9: The asterisk in (m * ,0) is weird.

We thank the reviewer for the comment. The asterisk is used to specify that from all potential donors, the most suitable match is chosen to be the donor for the specific recipient. We modified the line to clarify this in the context on page 5, line 25.

*"...which means, the selected donor, noted with asterisk (m*,0), is closest to the recipient and has sufficiently similar radiances."*

- Page 5, Line 12: 'of construction' → 'of the construction'.

The sentence has been changed as suggested.

- Page 6, Eq 5: Inconsistent use of italics. MR should not be italicized (use \mathrm) and N should be italicized. Note other usages of MR below.

The equation has been changed, as well as the usages of MR in the explanation.

$$\mathrm{MR} = (\mathrm{Agree}_{cr} + \mathrm{Agree}_{cd} + \mathrm{Agree}_{ae}) / N , \tag{5}$$

- Page 6, Line 30: 'km AGL' → 'km above AGL'.

Since the AGL stands for above ground level, we think no change is needed here.

- Page 7, Line 15: 'donors meets' → 'donors that meet'.

The sentence has been changed as suggested.

- Page 7, Line 18: 'air column' → 'air columns'.

The sentence has been changed as suggested.

- Page 7, Line 19: 'air column' → 'air columns'.

The sentence has been changed as suggested.

- Page 7, Line 29: 'This is resulted' → 'This results'.

The sentence has been changed as suggested.

- Page 7, Line 29: 'of TBM' → 'of the TBM'.

The sentence has been changed as suggested.

- Page 7, Line 32: 'by total' → 'by the total'.

The sentence has been changed as suggested.

- Page 7, Line 33: 'in reconstructed' → 'in the reconstructed'.

The sentence has been changed as suggested.

- Page 8, Line 1: 'in reconstructed' → 'in the reconstructed'.

The sentence has been changed as suggested.

- Page 8, Line 11: Use LaTex \times instead of 'x'.

The manuscript is not prepared in LaTex. We apologize for the inconvenience.

- Page 8, Line 16: 'In reality, this' → 'This'.

The sentence has been changed as suggested.

- Page 8, Line 29: Use LaTex \times instead of 'x'.

The manuscript is not prepared in LaTex. We apologize for the inconvenience.

- Page 8, Line 30: 'surrounding Bohai' → 'surrounding the Bohai'.

The sentence has been changed as suggested, as well as other sentences with the same issue.

- Page 9, Line 13: 'heating. Average' → 'heating. The average'.

The sentence has been changed as suggested.

- Page 9, Line 16: 'C, average' → 'C, the average'.

The sentence has been changed as suggested.

- Page 9, Line 20: 'AOD, especially' → 'AOD, with especially'.

The sentence has been changed as suggested.

- Page 9, Line 26: 'with CALIPSO' → 'with the CALIPSO'.

The sentence has been changed as suggested.

- Page 9, Line 27: 'hour of' → 'hours of'.

The sentence has been changed as suggested.

- Page 9, Line 27: 'using 10' → 'using the 10'.

The sentence has been changed as suggested.

- Page 9, Line 31: 'have larger' → 'have a larger'.

The sentence has been changed as suggested.

- Page 10, Line 9: 'to alter our' → 'to improve our'.

The sentence has been changed as suggested.

- Page 10, Line 10: 'for study' → 'for the study'.

The sentence has been changed as suggested.

- Page 10, Line 12: 'using SRM' → 'using the SRM'.

The sentence has been changed as suggested, as well as other sentences with the same issue.

- Page 10, Line 16: '6.%' → '6%'.

The extra period has been removed.

- Page 10, Line 18: 'with sufficient' → 'with a sufficient'.

The sentence has been changed as suggested.

- Page 10, Line 25: 'on SRM' → 'on the SRM'.

The sentence has been changed as suggested.

- Page 10, Line 25: 'a power' → 'an important'.

The sentence has been changed as suggested.

- Page 10, Line 26: 'well off' → 'not'.

The sentence has been changed as suggested.

- Page 16, Table 3: Identify the abbreviations in the headers.

Thank for the suggestion, we add the following footnote to the Table 3.

"'*No Sig' stands for 'no signal' and 'Surf' stands for 'surface and subsurface' portions of the measured columns."

- Page 20, Figure 3: 'with 30km' → 'with a 30km'.
- Page 20, Figure 3: \sigma should be in italicized.

We thank reviewer for the suggestion. After considering other review's comments, Figure 3 has been removed from the manuscript and the order of figures has been changed accordingly.

Anonymous Referee #2

The authors use the cloud construction algorithm developed by Barker et al. to construct vertical profile of aerosols. The algorithm seems to be exactly the same as the algorithm of Barker et al. with a different spatial resolution (a 5 km resolution for the lidar instrument). The study is focused on aerosol vertical profiles. They construct a 201 km wide of global aerosol profiles over two time periods. They test the algorithm by the same method discussed by Barker et al. They also evaluate the aerosol optical depth computed over nadir and off-nadir view regions and compare with the aerosol optical depth derived from AERONET.

I have serious concerns on this version. First, the algorithm developed by Barker et al. uses near infrared and infrared channels. I expect that the algorithm does not work for aerosols for two reasons. Aerosol signals in near IR and IR channels are very weak and I am surprised that the algorithm works for aerosols. In addition, aerosol signal in the upward radiance is very small over land. The variability of visible radiance over land is dominated by the variability of surface reflectance. Again, I expect that the Barker algorithm does not work for aerosols especially over land. These lead me to wonder why the algorithm works. There is no physical explanation provided in the manuscript. The authors need to explain reasons why the authors expect that the Barker algorithm works for aerosols. The current manuscript is written in a way that the authors took the algorithm and found it works for aerosols.

We would like to thank the reviewer for the questions.

The algorithm we used, developed by Barker et al.(2011), utilized 0.62–0.67, 2.105–2.155, 8.4–8.7, and 11.77–12.27 μm channels from MODIS, or simply denoted as bands 1,7,29 and 32. Among these bands, bands 1 and 7 are also used for MODIS aerosol retrieval. Therefore, we believe the algorithm can also work for aerosols, as assumed in Barker et al.(2011). We did notice that the choice of channels might not be as beneficial for aerosols as they are for clouds, and the sensitivity to aerosols might rely more on using visible channels, and the signal from aerosols could be much weaker.

We had done the following test in order to test and possibly optimize the algorithm for application on aerosols. We collected 30 days of CALIPSO profiles in 2015 with clear-sky condition and heavy aerosol loading at the east coast of China. We expected the manually selected cloudless dataset with heavy loading events would give a clear answer to whether the algorithm could work for aerosols or not. We

test the following combination of radiance bands: 1) using bands 1, 7, 29 and 32 used by Barker et al.(2011); 2) using bands 1 and 7 only; 3) using visible bands 1, 2, 3 and 4. We tested the performance of using these combinations by reconstructing the profile with dead-zone setting for 30 and 100 km.

[Figure]

**Figure AC1** Reconstruction of CALIPSO profile passing the east coast of China on 3 January 2015 with dead-zone setting for 100 km.

A typical result is shown in the following figure, where the panel a is the original profile, panel b to d corresponds to combination 1 to 3, panel e shows the results of choosing the closest pixels outside the dead-zone. The results we got from this test indicate that the original combination used by Barker et al.(2011) could get a pretty successful reconstruction (on average 81.9% and 75.2% matching rate at 30 and 100 km, respectively), which means it can be used to construct aerosol vertical structure. In comparison, using visible channels only have lower matching rate (around 60-70%), especially when aloft aerosol layer are present. The closest pixel method, on the other hand, has very high matching rate at 30 km, but as the dead-zone range increases or if the aerosol layer is not continuous, the simple horizontal shift leads to more errors.

In conclusion, based on the test results with heavy aerosol loading events, the algorithm works for aerosols with the channels selected in Barker et al.(2011). Since the decision is to use the original method, and to keep the method section concise, we decide not to include this part in the main text. On the other hand, we have added following brief explanation to the text on page 5, lines 14-16:

*"The bands are chosen for their widely accepted usage in retrieving cloud properties, including cloud cover, cloud top properties (CTP/CTT/CTH), and cloud phase (Ackerman et al., 1998;Baum et al., 2012;Baum et al., 2000) as well as aerosol properties (Sayer et al., 2014;Levy et al., 2013;Remer et al., 2013)."*

Second, the authors compare the aerosol optical depth for validation of, essentially, the Barker's algorithm. Figure 7 clearly shows that MODIS AODs agree with AERONET AOD better than "nadir" and "expanded". MODIS instruments have a full coverage of Earth. Therefore, comparison of AOD presented in the paper does not show any advantage of the aerosol-constructing algorithm compared to MODIS. The authors need to show the validation of constructed aerosol vertical profiles, perhaps compared with ground-based observations to demonstrate the advantage of constructing aerosol profiles. The authors also need to show how the error in the vertical aerosol profiles reduces, compared with nadir view only, when constructed aerosol profiles are used.

For reviewer's second concern, we want to clarify that we did not intend to get a 'better' quantification of AOD by expanding active profiles to nearby regions. After all, the active pixel being matched to the passive column could only provide an estimate of the column condition, which is not expected to be better than actual measurements MODIS made there. The advantage of this algorithm is really to help infer a profile and related vertical information, as shown in Figure 2 and original Figure 6 (now Figure 5), which passive-only cannot obtain.

We have added and modified the lines (page 12, lines 17-20) in Section 5 to illustrate this point.

*"The method in this work is not intended to get a precise quantification of aerosol profile, but to provide an estimate of the column's vertical structure. We did expect, to some extent, the estimation could be improved through calculations with constrains such as the column AOD measured by passive sensor at the exact location of the recipient pixel, which will need a lot more work in the future. In addition, since lacking of more suitable pixels is responsible for about half of the mismatching results, we are looking forward to launching more satellites with active and passive sensors, and possibly combining data from multiple satellite systems."*

As for reviewer's suggestion to validate the constructed aerosol vertical profiles with ground-based observations, given the difficulty of evaluating the aerosol vertical structure (AVS) retrieval due to scarcity of AVS measurements, in the original manuscript, we took one step back and compared the column total with ground-based AERONET sites.

Thanks to reviewer's comment and also same comment from reviewer #2, we made our best effort to find measurements we were able to compare with. We found the Asian dust and aerosol lidar observation network (AD-Net) when we looked for ground-based lidar stations with freely downloadable data. AD-Net is a lidar network for continuous observations of vertical distributions of Asian dust and other aerosols in East Asia. The sites contribute to the WMO GAW Program, and form the East Asian component of the GAW Aerosol Lidar Observation Network (GALION). Although cooperative stations in China didn't provide data sharing, we found some data that we were able to compare at Seoul station (37.5N,127.0E).

Seoul station has a standard lidar system in AD-Net, which is a two-wavelength (1064 nm, 532 nm) polarization sensitive (532 nm) Mie-scattering lidar, plus a 532 nm Raman (Shimizu et al. 2004). Based on the ground track, the A-Train sensors made overpass near the station for a total of 6 days during our case study in spring 2015. However, 4 out of these 6 days were heavily cloudy. For the remaining 2 days, March 7th and April 24th, the comparisons among ground-based lidar profiles, CALIPSO profiles at shortest distance and RXS-expand profiles averaged 25 km around the location of Seoul station are shown in the following figure, which is the new Figure 6 in the revised manuscript.

[Figure]

**Figure 6** Comparisons among ground-based lidar profiles of 532 nm attenuated backscatter coefficient products (units: m$^{-1}$sr$^{-1}$, averaged 2 hours within satellite overpass), CALIPSO profiles at shortest distance and RXS-expand profiles averaged

25 km around the location of Seoul station. The two plots on the left are from 7 March 2015, and the two plots on the right are from 24 April 2015.

The CALIPSO measurements we used for comparisons are level 1.5 data product of attenuated backscatter profiles, which clouds, overcast, surface, subsurface, and totally attenuated samples have been removed before being averaged to a 20 km horizontal resolution. In this case, RXS-expand profiles are based on the same products. The ground-based measurements used for comparison are the 532 nm attenuated backscatter coefficient products, averaged within 2 hr before and after the satellite overpass with 15 min time resolution.

For the aerosol layer 0-4 km above the ground, the relative error between CALIPSO profiles and ground station profiles are on average 21.6% on March 7th, and 18.7% on April 24th. The distances between station and ground track are 51.0 km on the first day, and 50.1 km on the second. Between RXS-expand profiles and ground station profiles, the average relative error is 27.9% and 23.4%, respectively.

For the aerosol layer 0-4 km above the ground, the relative error between CALIPSO profiles and ground station profiles are on average 21.6% on March 7th, and 18.7% on April 24th. The distances between station and ground track are 51.0 km on the first day, and 50.1 km on the second. Between RXS-expand profiles and ground station profiles, the average relative error is 27.9% and 23.4%, respectively. The results from the comparisons agreed in general. Previous studies found that there were considerable disagreement between CALIPSO measurements and ground-based lidar measurements; in most studies, the differences were found to be around 20% (Mamouri et al., 2009;Wu et al., 2011;Kim et al., 2008;Chiang et al., 2011). For example, Mamouri et al. (2009) compared CALIPSO attenuated backscatter coefficient profiles with a ground-based lidar in Athens, Greece, and they found the agreement on the order of −10±12% for cloud-free daytime measurements between 3 and 10 km, while the differences between 1 and 3 km were much larger (−34±34%).

We made the decision to add following lines to Section 4.3, on page 10-11, as well as a few related sentences in the abstract and summary.

"In the three-month period, the Asian dust and aerosol lidar observation network (AD-Net) site at Seoul, Korea (37.5N,127.0E) provided measurements of atmospheric profiles that we were able to compare with those constructed in the surrounding area using the SRM method. Seoul station has a

*standard lidar system in AD-Net, which is a two-wavelength (1064 nm, 532 nm) polarization sensitive (532 nm) Mie-scattering lidar, plus a 532 nm Raman (Shimizu et al., 2004). Based on the ground track, the A-Train sensors made overpass near the station for a total of 6 days during that spring. However, 4 out of these 6 days were heavily cloudy. For the remaining 2 days, 7 March and 24 April, the comparisons among ground-based lidar profiles, CALIPSO profiles at shortest distance and RXS-expand profiles averaged 25 km around the location of Seoul station are shown Fig. 6.*

*The CALIPSO measurements used for comparisons are level 1.5 data products of attenuated backscatter profiles, which clouds, overcast, surface, subsurface, and totally attenuated samples have been removed before being averaged to a 20 km horizontal resolution. In this case, RXS-expand profiles are based on the same products. The ground-based measurements used for comparison are the 532 nm attenuated aerosol backscatter coefficient products, averaged within 2 hr before and after the satellite overpass with 15 min time resolution.*

*For the aerosol layer 0-4 km above the ground, the relative error between CALIPSO profiles and ground station profiles are on average 21.6% on 7 March, and 18.7% on 24 April. The distances between station and ground track are 51.0 km on the first day, and 50.1 km on the second. Between RXS-expand profiles and ground station profiles, the average relative errors are 27.9% and 23.4%, respectively. The results from the comparisons agreed in general. Previous studies found that there were considerable disagreement between CALIPSO measurements and ground-based lidar measurements; in most studies, the differences were found to be around 20% (Mamouri et al., 2009; Wu et al., 2011; Kim et al., 2008; Chiang et al., 2011)."*

Anonymous Referee #3

Aerosol vertical structure (AVS) determines much of the climate impact of aerosol in the atmosphere, but it is difficult to measure with the spatial and temporal coverage needed for many applications. This study uses the spectral radiance matching (SRM) method, which infers AVS from column measurements by matching them to similar measurements with collocated vertical profiles, to construct global AVS from MODIS and CALIOP data. The paper is well organized, and the methods section is especially clearly written.

Although collocated AERONET data is widely used to validate MODIS aerosol, its use in a case study for this manuscript does not seem sufficient to evaluate the AVS retrieval. This is undoubtedly difficult, because the scarcity of AVS data that makes this study so valuable also leaves little basis for comparison. However, airborne field campaigns and ground-based lidar networks do measure vertical profiles of clouds and aerosol on a smaller scale, and given the global MODIS/CALIOP record to choose from, it should be possible to use some of these measurements to validate the satellite AVS.

We thank reviewer's understanding of the difficulty of evaluating the AVS retrieval due to scarcity of AVS measurements, and we made our best effort to find measurements we were able to compare with.

We found the Asian dust and aerosol lidar observation network (AD-Net) when we looked for ground-based lidar stations with freely downloadable data. AD-Net is a lidar network for continuous observations of vertical distributions of Asian dust and other aerosols in East Asia. The sites contribute to the WMO GAW Program, and form the East Asian component of the GAW Aerosol Lidar Observation Network (GALION). Although cooperative stations in China didn't provide data sharing, we found some data that we were able to compare at Seoul station (37.5N,127.0E).

Seoul station has a standard lidar system in AD-Net, which is a two-wavelength (1064 nm, 532 nm) polarization sensitive (532 nm) Mie-scattering lidar, plus a 532 nm Raman (Shimizu et al. 2004). Based on the ground track, the A-Train sensors made overpass near the station for a total of 6 days during our case study in spring 2015. However, 4 out of these 6 days were heavily cloudy. For the remaining 2 days, March 7th and April 24th, the comparisons among ground-based lidar profiles, CALIPSO profiles at shortest distance and RXS-expand profiles averaged 25 km around the location of Seoul station are shown in the following figure, which is the new Figure 6 in the revised manuscript.

[Figure]

**Figure 6** Comparisons among ground-based lidar profiles of 532 nm attenuated backscatter coefficient products (units: m$^{-1}$sr$^{-1}$, averaged 2 hours within satellite overpass), CALIPSO profiles at shortest distance and RXS-expand profiles averaged 25 km around the location of Seoul station. The two plots on the left are from 7 March 2015, and the two plots on the right are from 24 April 2015.

The CALIPSO measurements we used for comparisons are level 1.5 data product of attenuated backscatter profiles, which clouds, overcast, surface, subsurface, and totally attenuated samples have been removed before being averaged to a 20 km horizontal resolution. In this case, RXS-expand profiles are based on the same products. The ground-based measurements used for comparison are the 532 nm attenuated backscatter coefficient products, averaged within 2 hr before and after the satellite overpass with 15 min time resolution.

For the aerosol layer 0-4 km above the ground, the relative error between CALIPSO profiles and ground station profiles are on average 21.6% on March 7th, and 18.7% on April 24th. The distances between station and ground track are 51.0 km on the first day, and 50.1 km on the second. Between RXS-expand profiles and ground station profiles, the average relative error is 27.9% and 23.4%, respectively.

For the aerosol layer 0-4 km above the ground, the relative error between CALIPSO profiles and ground station profiles are on average 21.6% on March 7th, and 18.7% on April 24th. The distances between station and ground track are 51.0 km on the first day, and 50.1 km on the second. Between RXS-expand profiles and ground station profiles, the average relative error is 27.9% and 23.4%, respectively. The results from the comparisons agreed in general. Previous studies found that there were considerable disagreement between CALIPSO measurements and ground-based lidar measurements; in most studies,

the differences were found to be around 20% (Mamouri et al., 2009;Wu et al., 2011;Kim et al., 2008;Chiang et al., 2011). For example, Mamouri et al. (2009) compared CALIPSO attenuated backscatter coefficient profiles with a ground-based lidar in Athens, Greece, and they found the agreement on the order of −10±12% for cloud-free daytime measurements between 3 and 10 km, while the differences between 1 and 3 km were much larger (−34±34%).

In addition, we want to clarify that we did not intend to get a precise quantification of aerosol profile through the scene construction method. After all, the active pixel being matched to the passive column is intended only to provide an estimate of the column's vertical structure. We did expect, to some extent, the estimation could be improved through calculations with constrains such as the column AOD measured by passive sensor at the exact location of the recipient pixel, which will need a lot more work in the future.

We made the decision to add following lines to Section 4.3, on page 10-11, as well as a few related sentences in the abstract and summary.

*"In the three-month period, the Asian dust and aerosol lidar observation network (AD-Net) site at Seoul, Korea (37.5N,127.0E) provided measurements of atmospheric profiles that we were able to compare with those constructed in the surrounding area using the SRM method. Seoul station has a standard lidar system in AD-Net, which is a two-wavelength (1064 nm, 532 nm) polarization sensitive (532 nm) Mie-scattering lidar, plus a 532 nm Raman (Shimizu et al., 2004). Based on the ground track, the A-Train sensors made overpass near the station for a total of 6 days during that spring. However, 4 out of these 6 days were heavily cloudy. For the remaining 2 days, 7 March and 24 April, the comparisons among ground-based lidar profiles, CALIPSO profiles at shortest distance and RXS-expand profiles averaged 25 km around the location of Seoul station are shown Fig. 6.*

*The CALIPSO measurements used for comparisons are level 1.5 data products of attenuated backscatter profiles, which clouds, overcast, surface, subsurface, and totally attenuated samples have been removed before being averaged to a 20 km horizontal resolution. In this case, RXS-expand profiles are based on the same products. The ground-based measurements used for comparison are the 532 nm attenuated aerosol backscatter coefficient products, averaged within 2 hr before and after the satellite overpass with 15 min time resolution.*

*For the aerosol layer 0-4 km above the ground, the relative error between CALIPSO profiles and ground station profiles are on average 21.6% on 7 March, and 18.7% on 24 April. The distances between station and ground track are 51.0 km on the first day, and 50.1 km on the second. Between RXS-expand profiles and ground station profiles, the average relative errors are 27.9% and 23.4%, respectively. The results from the comparisons agreed in general. Previous studies found that there were considerable disagreement between CALIPSO measurements and ground-based lidar measurements; in most studies, the differences were found to be around 20% (Mamouri et al., 2009; Wu et al., 2011; Kim et al., 2008; Chiang et al., 2011)."*

We also added and modified the lines (page 12, lines 17-20) in Section 5 for clarification of the purpose of the SRM method.

*"The method in this work is not intended to get a precise quantification of aerosol profile, but to provide an estimate of the column's vertical structure. We did expect, to some extent, the estimation could be improved through calculations with constrains such as the column AOD measured by passive sensor at the exact location of the recipient pixel, which will need a lot more work in the future. In addition, since lacking of more suitable pixels is responsible for about half of the mismatching results, we are looking forward to launching more satellites with active and passive sensors, and possibly combining data from multiple satellite systems."*

Specific comments:

Page 2, lines 1-23. I'm surprised not to see any mention here of ground-based lidar networks, which are sometimes used in combination with CALIOP data; or of the shorter-lived NASA CATS lidar that was aboard the ISS.

We thank the reviewer for the suggestion and we added the following lines to the content, as a separate paragraph after the first one on page 2, lines 16-20.

*"The development of lidar technology helped provide these vital missing piece of information. Ground-based lidar systems have been stationed at various locations and also used in field campaigns to measure the vertical and horizontal distribution of aerosols (Welton et al., 2000; Welton et al., 2002; Badarinath et al., 2010). Ground-based lidars provide measurements on the fixed locations on timescale of minutes to hours, depending on the specific type of lidar used in the experiment. Limited by the*

*stationary setting, ground-based lidars could not achieve true global coverage, nevertheless, network of ground-based lidars (e.g. MPL-NET, EARLINET, AD-NET) provide key insights to atmospheric study and are involved in validation of satellite sensors (Kovacs et al., 2004; Mamouri et al., 2009; Pappalardo et al., 2010)."*

Page 6, lines 18-21. These cloud cover rates seem very low. For passive sensors, 70% is a reasonable ballpark estimate for the fraction of the globe covered by clouds at any given time. Most such clouds would occupy only a small part of the vertical column (and as the paper states, almost never at high altitudes) but the numbers still seem difficult to reconcile. Have you calculated the global cloud cover from the column perspective, for comparison?

We thank the author for the question and comment. We made it more clear in the revision that the numbers in Table 2 refers to cloud occurrence as percentage they occupied in the vertical column. We calculated this occurrence rate according to CALIPSO VFM products, which was scaled for vertical and horizontal resolution (Hunt et al. 2009). However, it is true that the numbers in the table underestimated the amount of clouds in actual atmosphere. As we stated, CALIPSO's signal can be totally attenuated beneath clouds and possibly making cloud layers below showed up as "no signal". The horizontal cloud coverage between 60°N and 60°S for the tested periods in April and September 2015 are 68.7% and 71.3%, respectively.

We added and modified the lines on page 7, lines 7-13 to address this comment.

*"Table 2 summarizes frequencies of occurrences of atmosphere conditions. These numbers refer to the occurrence of atmospheric features as percentage they occupied in the vertical column. We calculated this occurrence rate according to CALIPSO VFM products, which was then scaled for vertical and horizontal resolution of the products (Hunt et al., 2009).The majority of conditions were identified as clear. Above 8.2 km clear arrays occur over 90% of the time. Aerosols and clouds occurred below 8.2 km, in about 7% and 5% of the cells, respectively. Note that arrays identified as 'no signal' represent 16 - 18% of cells in this layer; CALIPSO's signal can be totally attenuated beneath opaque clouds and certain aerosols. This indicates that the numbers in Table 2 likely underestimated the amount of clouds and aerosols in actual atmosphere."*

Page 6, Figure 2. This is fascinating. It would be interesting to see a more detailed discussion of the contrast between 0-2 km and 2-4 km, which appear to distinguish local aerosol from aerosol undergoing long-range transport.

We thank the author for the suggestion. We add more discussion about the contrast between 0-2 km and 2-4 km into the last paragraph on page 7, lines 21-31.

*"Height-resolved global AOD maps (averaged for a 2 °× 2 °lat/long grid) based on the two selected periods are shown in Fig. 2. In the near-surface layer, 2 km above ground level (AGL), in April, relatively high aerosol loadings are found in the cross-Atlantic African dust transport, Saudi Arabia, and India. In September, dust dynamics are much weaker but much biomass burning is apparent in the Brazilian Amazon and Southern Africa. This seasonal trend of dust and smoke is more obvious in the layer 2-4 km AGL. Aerosol in this layer aloft are expected to be undergoing long-range transport. In April, the thickest dust layers are found slightly inland of the western coast of Africa, around 12.5 °N, 5.5 °E, and in the centre of Saudi Arabia around 24.5 °N, 42.5 °E. The shift of AOD distribution between surface layer and layer above is logical, and indicates the movement of dust layers as the aerosol loadings are transported towards the oceans. In September, this contrast is harder to observe as the dust dynamic is weaker, but similar trends are found in the biomass burning regions. In addition, persistent high aerosol loadings in both 0-2 km and 2-4 km AGL are found in India and the east coast of China with mixed sources of natural aerosols and pollutants. The results could be affected by the local topography."*

Page 8, Figure 3. This plot is somewhat difficult to read. A different color scheme may make the drop in the matching rate at the ITCZ easier to spot, but I'm having trouble seeing any other patterns.

We agree with the reviewer that Figure.3 is rather vague for the information we tried to convey. In fact, we think Figure.3 does not give enough extra information on its own, other than these results given in Figure.4. Therefore, we make a decision to remove this figure and related sentences, change the order of figures accordingly.

Technical comments:

Page 6, line 14. "Losing".

Page 8, line 19. "CALIOP profiles".

The sentence has been changed as suggested.

[revised manuscript text omitted]

---

## Author Response (AR2)

Dear Editor Andrew,

We would like to thank you and all the reviewers again for the time spent to re-evaluate the paper and for the suggestions you gave which help to improve our work. Here we submit a final version of manuscript which includes all adjustments requested by your comments. We also fixed a missing label on Figure.5 during our last check of manuscript for typos and other problems. Thanks again for your help.

"The authors reply to my first major concern well. Although the authors indicate that they do not include the material in the paper, I suggest adding one section and including this result in the paper. The title of the section would be "testing selection of wavelengths for aerosol applications" or something similar and discuss how the selection of wavelengths made by Barker et al. performs for aerosols."

On the first point, you had performed the test but omitted the results from the manuscript to keep the method selection concise; I appreciate the additional effort here. However, I agree with the referee that there is value in adding this information briefly into the revised manuscript), in case other readers wonder about wavelength selection. I do not think this would make that part of the paper too long.

We thank the editor and reviewer for the suggestion. The following content has been added to the manuscript Section 3 as a separate subsection (new Section 3.2). We also modified a couple sentences in Section 3.1 referring to the new content.

*"Since the construction algorithm is initially developed for clouds, efforts have been made to apply the algorithm to aerosols. The following test is performed to find the possible combination of bands most sensitive to aerosols. 30 days of CALIPSO profiles at the east coast of China in 2015 are selected and screened with clear-sky and heavy aerosol loadings conditions. The manually selected cloudless datasets with heavy loading events are expected to give a clear indication of whether the algorithm could work for aerosols or not. The following combinations of radiance bands are tested: 1) a combination of bands 1, 7, 29 and 32 used by Barker et al. (2011); 2) a combination of bands 1 and 7 only; 3) a combination of visible bands 1, 2, 3 and 4. The performance of algorithm using these combinations is evaluated by reconstructing the profile with dead zone setting for 30 and 100 km.*

*A typical comparison among the reconstructed profiles is shown in the Fig. 1, where the panel (a) is the original profile, panel (b) to (d) corresponds to combination 1 to 3 described above, panel (e) shows the reconstructed profile from directly choosing the closest pixels outside the dead zone. The results of the test indicate that the bands used by Barker et al. (2011) could get a pretty successful reconstruction. The matching rate at 30 and 100 km are on average 81.9% and 75.2%, respectively, which means this combination can be used to construct aerosol vertical structure. In contrast, using visible bands only have lower matching rate (around 60-70%), especially when aloft aerosol layer are present. Selecting the closest pixel, on the other hand, has very high matching rate at 30 km, which is expected since aerosol properties are relatively horizontally uniform. However, as the dead zone range increases or in cases that*

*aerosol layer is not continuous, the simple horizontal shift leads to more errors. Based on the test results with heavy aerosol loading events, the combination of bands for aerosol application is the same as the wavelength selection in Barker et al. (2011)."*

[Figure]

**Figure 1** Reconstruction of CALIPSO profile passing the east coast of China on 3 January 2015 with dead zone setting for 100 km. The panels show the original profile and reconstructed profiles using different combinations of radiance bands.

On the second point, this is perhaps in part a philosophical matter. My understanding from reading your paper and response is that while the AOD you get is not as good as MODIS standard, and the profile has some limitations compared to CALIOP standard, what you see as the main benefit is getting some profile information over a broader instantaneous swath than available from CALIOP alone. So, while both MODIS and CALIOP provide near-global coverage on a long-term basis, the important thing here is the gain in profile information on an instantaneous basis, which might be important for some instantaneous applications (e.g. feature tracking, air quality). Is that right? I suggest adding an additional sentence or two signposting this more clearly. I do not think it needs to be demonstrated on those applications in this paper, but direct statements would be useful.

*We thank the editor for the suggestion. Your understanding is correct. We have added the following sentences to clarify the significance of the work on page 3, lines 5-8.*

[revised manuscript text omitted]

